# Streaming Covariate Balancing via Discrepancy-Based Feature Coresets

**Yixin Ren**[1]  **Chenghou Jin**[1]  **Yewei Xia**[1]  **Zichuan Lin**[†2]  **Deheng Ye**[2]  **Hao Zhang**[3]
**Jihong Guan**[4]  **Shuigeng Zhou**[†1]

## Abstract

Real-time estimation of average treatment effects (ATE) in streaming observational data poses two key challenges: strict memory constraints that preclude storing the full data history, and distributional shifts in both treatment assignment and outcome-generating process. Existing methods either require offline access to the entire dataset for covariate balancing or rely on parametric online models that are vulnerable to model misspecification under such shifts. This paper proposes a novel model-agnostic method for ATE estimation in streaming data, which effectively addresses the above challenges. Based on discrepancy theory, we first compress streaming data into feature coresets that preserve covariate balancing objectives over a rich nonparametric function class, enabling linear-time updates with bounded memory. Then, by directly learning balancing weights and bypassing parametric propensity score estimation, we enhance the model's robustness against the shift in treatment assignment, while by balancing over an expressive function space we make the model more adaptive to the shift in the outcome-generating process. Theoretically, we establish convergence guarantees with explicit bounds on memory usage and computational complexity. Empirically, extensive experiments on both synthetic and real-world datasets show the effectiveness and robustness of the proposed method, consistently outperforming existing techniques.

[1]College of Computer Science snd Artificial Intelligence, Fudan University, Shanghai, China [2]Tencent Hunyuan [3]Shenzhen Institutes of Advanced Technology, Chinese Academy of Sciences, Shenzhen, China [4]School of Computer Science and Technology, Tongji University, Shanghai, China. Correspondence to: Zichuan Lin <lzcthu12@gmail.com>, Shuigeng Zhou <sgzhou@fudan.edu.cn>.

*Proceedings of the $43^{rd}$ International Conference on Machine Learning*, Seoul, South Korea. PMLR 306, 2026. Copyright 2026 by the author(s).

## 1. Introduction

Causal inference from observational data seeks to estimate treatment effects when randomized controlled trials (RCTs) are unavailable or impractical (Concato et al., 2017). In such settings, treatment assignment is not randomized, and treated and control units may differ systematically in their covariate distributions, leading to confounding (Shalit et al., 2017). In offline settings with access to the full dataset, a rich literature has developed techniques for estimating average treatment effects (ATE). Broadly, existing approaches can be categorized into *model-based methods* (Hirano et al., 2003; Rosenbaum & Rubin, 1983), which rely on parametric models for the propensity score and/or outcome regression, and *discrepancy-based covariate balancing methods* (Yan et al., 2024; Kong et al., 2023), which directly reweight samples to balance covariate distributions between treated and control groups with respect to a function-space discrepancy.

In real-world causal inference applications, data often arrive sequentially over time, and streaming implementations typically operate under tight storage and per-update computational budgets (Lin et al., 2023). Moreover, the data-generating process is often non-stationary and exhibits distributional shifts (Chen et al., 2025). A representative example is vaccine safety surveillance (Lieu et al., 2007; Luo et al., 2024), where routinely collected healthcare data accumulate continuously, making repeated storage or reprocessing of the full data history prohibitively expensive. In this scenario, vaccination policies may induce shifts in the treatment assignment mechanism, while changes in disease dynamics can alter the outcome-generating process. These challenges motivate streaming ATE estimation under limited memory and computation in the presence of distributional shifts.

Recent works on streaming causal inference have primarily focused on developing online estimators for classical ATE methods. For example, Luo et al. (2024) formulates IPTW, G-comp, and AIPTW as M-estimators and updates them sequentially using batch-level summary statistics, thereby reducing storage and computational cost. Under regularity conditions and correct nuisance model specification, their estimators are asymptotically equivalent to the pooled-data oracle. However, such model-based approaches still rely on pre-specified parametric propensity score and outcome

regression models, which can be fragile under distributional shifts: changes in treatment assignment may invalidate propensity score estimates and lead to unstable weighting, while shifts in outcome generation can degrade outcome regression accuracy, undermining regression-based estimators. These limitations motivate model-agnostic alternatives that remain computationally efficient and performance-stable under evolving data distributions.

To overcome memory constraints and enhance robustness to distributional shifts in existing model-based online estimators, we propose a novel model-agnostic method for ATE estimation in streaming data. Building on discrepancy theory, we first compress streaming data into feature coresets that preserve covariate balancing objectives over a rich nonparametric function class, enabling linear-time updates with bounded memory. Then, by directly learning balancing weights without estimating a parametric propensity score, our method reduces the model's sensitivity to shifts in the treatment assignment mechanism, while balancing over an expressive function space to enable the model to adapt to shifts in the outcome-generating process. The method processes streaming samples in linear time with respect to the number of observations $n$ and requires bounded memory on the order of $\mathcal{O}(\log n/\varepsilon^2)$, where $\varepsilon$ controls the additional approximation error in ATE estimation.

In summary, our contributions are as follows: 1) We introduce a model-agnostic method for average treatment effect (ATE) estimation in streaming data, designed to operate under strict memory constraints and distributional shifts. 2) We propose *Feature Coreset*, a mergeable data summary that provably preserves the key statistics required for accurate weighted ATE estimation, together with an efficient online weight learning procedure. 3) We establish end-to-end theoretical guarantees for the proposed method, including explicit bounds on estimation error, memory usage, and computational complexity. 4) We empirically demonstrate the effectiveness, scalability, and robustness of our method on both synthetic and real-world datasets.

## 2. Related Work

**Causal effect estimation.** Estimating causal effects from observational data is challenging due to confounding and covariate imbalance between treated and control groups. A classical approach is inverse probability weighting (IPW), which reweights samples by the inverse of their propensity scores (Hirano et al., 2003; Rosenbaum & Rubin, 1983). In practice, propensity scores must be estimated (e.g., via logistic regression or more flexible models) (Lunceford & Davidian, 2004; Lee et al., 2010), and the resulting weights can be sensitive to estimation errors, leading to unstable and biased effect estimates (Li et al., 2018; Hainmueller, 2012; Imai & Ratkovic, 2014). Doubly robust estimators mitigate

this issue by combining reweighting with outcome regression, remaining consistent if either the propensity model or the outcome model is correctly specified (Robins et al., 1994). An alternative, model-agnostic direction is to directly learn weights that balance covariate distributions across treatment groups, without explicitly modeling the propensity score (Kuang et al., 2017). More generally, covariate balancing can be formulated as minimizing a discrepancy between weighted treated and control covariate distributions over a function class, including objectives based on Wasserstein distances (Yan et al., 2025; 2024; Villani et al., 2008) and kernel-based discrepancies (Kong et al., 2023). To further enhance expressiveness, representation learning has been used to map covariates into a feature space where treated and control distributions are easier to align, improving treatment effect estimation in complex settings (Li & Fu, 2017; Kallus, 2020; Shalit et al., 2017; Johansson et al., 2016; Assaad et al., 2021; Johansson et al., 2022).

Most existing methods are designed for offline settings with full data access. Differently, this paper studies streaming ATE estimation under fixed memory budgets and evolving data distributions, where sequential nuisance modeling can be brittle under temporal shifts. Our method leverages a lightweight randomized feature representation to enable rich-function covariate balancing in a streaming setting.

**Coresets.** Coresets provide compact summaries of large datasets with provable approximation guarantees and have been widely used in scalable learning and inference (Campbell & Broderick, 2019; Cohen-Addad et al., 2022), including distributional objectives such as Wasserstein measure coresets (Claici et al., 2018). A key theoretical tool underlying many coreset constructions is *discrepancy*, which measures how well a weighted subset approximates the full dataset with respect to a target class of test functions. This perspective has led to substantially improved guarantees over naive sampling in a variety of settings (Phillips & Tai, 2018; 2020; Karnin & Liberty, 2019), and has been further advanced by recent progress in algorithmic discrepancy minimization (Chazelle, 1998; Alweiss et al., 2021; Kulkarni et al., 2024). As a representative application, discrepancy-based techniques yield near-optimal coresets for kernel density estimation (KDE), including strong results for Gaussian KDE (Phillips, 2013; Tai, 2020a;b; 2022).

Despite these advances, existing discrepancy-driven coresets are mainly designed for generic approximation tasks and do not directly target causal estimands such as the ATE in streaming settings. In contrast, this work develops a task-aware, mergeable coreset construction tailored to online weighted ATE estimation. Our method uses lightweight randomized feature projections to enable covariate balancing over a broad nonparametric function space, while supporting efficient streaming updates with fixed memory.

## 3. Problem Formulation and Notation

We study causal covariate balancing in a streaming setting, where observations arrive sequentially over time. At time $i = 1, 2, \ldots$, we observe one unit $(\boldsymbol{x}_i, a_i, y_i)$, where $\boldsymbol{x}_i \in \mathcal{X} \subset \mathbb{R}^p$ denotes covariates, $a_i \in \{0, 1\}$ is the treatment indicator, and $y_i$ is the observed outcome. Given $\boldsymbol{x}_i$, treatment is assigned as $a_i \sim \text{Ber}(\pi(\boldsymbol{x}_i))$, where $\pi(\boldsymbol{x})$ is the propensity score. We assume *strict overlap*: there exists $\eta > 0$ such that $\eta \leq \pi(\boldsymbol{x}) \leq 1 - \eta$ for all $\boldsymbol{x} \in \mathcal{X}$.

Let $Y_0$ and $Y_1$ denote generic potential outcomes under control and treatment, respectively. We also assume the standard *ignorability* and *consistency* conditions (Rosenbaum & Rubin, 1983). Together with the strict overlap condition above, these assumptions allow the population causal effect to be identified from the observed data distribution.

**Streaming with shifts.** Let $\mathcal{D}^{(\tau)} = \{(\boldsymbol{x}_j, a_j, y_j)\}_{j=1}^{\tau}$ denote the data stream observed up to time $\tau$. As in standard offline causal inference, causal effects are defined with respect to a target population under an *i.i.d.* sampling framework. In the streaming setting, this sample is revealed sequentially rather than available as a single static dataset, and $\mathcal{D}^{(\tau)}$ denotes the observations available for estimation at time $\tau$. Our target estimand is the population ATE,

$$\text{ATE} := \mathbb{E}[Y_1] - \mathbb{E}[Y_0]. \tag{1}$$

Although the target estimand retains the same population-level formulation, the sequentially observed batches may exhibit changes in composition or nuisance mechanisms. We consider two such changes: *outcome shift*, where the outcome regression $r_a(\boldsymbol{x}) = \mathbb{E}[Y_a \mid \boldsymbol{X} = \boldsymbol{x}]$ for treatment level $a \in \{0, 1\}$ varies over time, and *treatment shift*, where $\pi(\boldsymbol{x})$ drifts over time. These shifts can make sequentially updated nuisance models inaccurate as the stream evolves.

**Weighted ATE estimator.** Let $\mathcal{I}_t := \{j \leq \tau : a_j = 1\}$ and $\mathcal{I}_c := \{j \leq \tau : a_j = 0\}$ be the treated and control indices, with sizes $n_t := |\mathcal{I}_t|$ and $n_c := |\mathcal{I}_c|$. Define the empirical covariate measures $\mu_t := \frac{1}{n_t} \sum_{j \in \mathcal{I}_t} \delta_{x_j}$, $\mu_c := \frac{1}{n_c} \sum_{j \in \mathcal{I}_c} \delta_{x_j}$, $\mu := \frac{1}{n_t + n_c} \sum_{j=1}^{\tau} \delta_{x_j}$. Given nonnegative balancing weights $\boldsymbol{b}_t \in \mathbb{R}_+^{n_t}$ and $\boldsymbol{b}_c \in \mathbb{R}_+^{n_c}$ satisfying $\mathbf{1}^\top \boldsymbol{b}_t = 1$, $\mathbf{1}^\top \boldsymbol{b}_c = 1$, we define the weighted covariate measures $\mu_{t,b} := \sum_{j \in \mathcal{I}_t} b_{t,j}\, \delta_{x_j}$, $\mu_{c,b} := \sum_{j \in \mathcal{I}_c} b_{c,j}\, \delta_{x_j}$. At each time $\tau$, our goal is to construct balancing weights $(\boldsymbol{b}_t, \boldsymbol{b}_c)$ that achieve *covariate balance*, meaning that the weighted treated and control covariate distributions are close to the target population covariate distribution, i.e., $\mu_{t,b} \approx \mu$ and $\mu_{c,b} \approx \mu$. We then estimate the ATE as follows:

$$\widehat{\text{ATE}} = \sum_{j \in \mathcal{I}_t} b_{t,j}\, y_j - \sum_{j \in \mathcal{I}_c} b_{c,j}\, y_j, \tag{2}$$

which targets the true population ATE in Eq. (1).

**Memory constraint and streaming objective.** A central challenge in the streaming setting is limited memory. The algorithm cannot store the full history $\mathcal{D}^{(\tau)}$; Instead, it maintains an online buffer of size $B_\tau$, so the memory footprint can grow over time while remaining far smaller than storing the full data stream. All computations, including coreset maintenance, weight learning, and ATE estimation, must be performed using only the buffered units. Our goal is to design a streaming algorithm that outputs an estimate $\widehat{\text{ATE}}_\tau$ at each time $\tau$ with provable accuracy guarantees.

**Notation.** For any measurable function $f$ and measure $\nu$, we use the shorthand inner product $\langle f, \nu \rangle := \int f(\boldsymbol{x})\, d\nu(\boldsymbol{x})$, which reduces to a weighted sum when $\nu$ is an empirical measure. We denote by $\Delta^{|\mathcal{S}|} := \{\boldsymbol{b} \in \mathbb{R}_+^{|\mathcal{S}|} : \mathbf{1}^\top \boldsymbol{b} = 1\}$ the probability simplex over a finite set $\mathcal{S}$. The notation $[m] := \{1, 2, \ldots, m\}$ denotes the index set of size $m$. Boldface symbols indicate vector- or matrix-valued quantities.

## 4. Methodology

We propose a model-agnostic method for average treatment effect (ATE) estimation in streaming data under a fixed memory budget. The method maintains compact, mergeable summaries of the data stream that preserve covariate-balancing objectives, enabling accurate weighted ATE estimation with efficient online updates. It combines random feature representations of a rich nonparametric function class with discrepancy-preserving streaming coresets, allowing covariate balancing to be performed directly on compressed data without relying on parametric models.

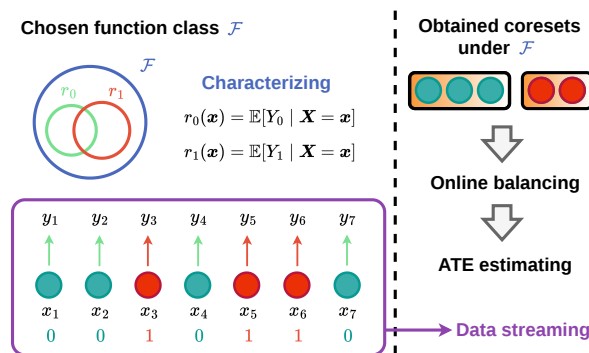

*Figure 1.* Overview of the proposed Feature Coreset (FCore) framework for memory-efficient streaming ATE estimation.

**Overview of FCore.** Figure 1 illustrates the overall workflow of the proposed FCore framework. We begin by choosing a function class $\mathcal{F}$ that characterizes the outcome regression functions $r_a(\boldsymbol{x}) = \mathbb{E}[Y_a \mid \boldsymbol{X} = \boldsymbol{x}]$, $a \in \{0, 1\}$. As observations arrive sequentially, FCore maps the covariates into a random feature space associated with $\mathcal{F}$ and maintains mergeable feature coresets for the treated and control

groups. These coresets provide compressed summaries of the streaming data while preserving the feature statistics needed for covariate balancing. Given the maintained coresets, FCore performs online balancing to learn weights that align the weighted treated and control summaries with the target population. The resulting weights are applied to the observed outcomes to produce the weighted ATE estimate.

**Choice of function class.** We consider outcome regression functions that belong to the integral feature class $\mathcal{F}$ defined in Assumption 4.1, which is standard in the random feature literature (Rahimi & Recht, 2008). This is a fairly broad and flexible class that, through suitable choices of feature maps and sampling distributions, can represent many smooth and nonlinear functions, including kernel-induced function classes. Balancing over this class therefore enables covariate balancing beyond linear specifications and helps accommodate temporal outcome shifts when the post-shift regression functions remain in the function class $\mathcal{F}$.

**Assumption 4.1** (Function class for outcome regression). The outcome regression functions $r_0$ and $r_1$ belong to the set $\mathcal{F} = \{f(\boldsymbol{x}) = \int_\Omega o(\boldsymbol{h})\psi(\boldsymbol{x};\boldsymbol{h})d\boldsymbol{h} \mid |o(\boldsymbol{h})| \leq Cp(\boldsymbol{h})\}$ where $\Omega$ is a measurable space, $p(\boldsymbol{h})$ is a probability density (or probability mass function) on $\Omega$ with respect to the base measure $d\boldsymbol{h}$, $C > 0$ is a constant and $\psi(\cdot;\boldsymbol{h}) : \mathcal{X} \to \mathbb{R}$ is a given measurable feature map. For technical simplicity, we assume $|\psi(\boldsymbol{x},\boldsymbol{h})| \leq 1$ for all $\boldsymbol{x}$ and $\boldsymbol{h}$.

Under Assumption 4.1, the integral feature representation admits a natural finite-dimensional approximation. Specifically, we draw $\boldsymbol{h}_1,\ldots,\boldsymbol{h}_d \overset{\text{i.i.d.}}{\sim} p(\boldsymbol{h})$ and define the $d$-dimensional random feature map as

$$\phi(\boldsymbol{x}) = \frac{1}{\sqrt{d}}\big(\psi(\boldsymbol{x};\boldsymbol{h}_1),\ldots,\psi(\boldsymbol{x};\boldsymbol{h}_d)\big)^\top \in \mathbb{R}^d. \quad (3)$$

This approximation replaces balancing over the infinite-dimensional class $\mathcal{F}$ with balancing in the induced finite-dimensional feature space, where $\|\phi(\boldsymbol{x})\|_2 \leq 1$.

### 4.1. Discrepancy-Based Feature Coresets Construction

Our method builds coresets that compress the data stream while preserving the statistics needed for covariate balancing and ATE estimation. This is inspired by the discrepancy-based framework of Karnin & Liberty (2019), which reduces uniform coreset approximation to a $\{\pm 1\}$ signing problem. Consider objectives of the form $F(\boldsymbol{q}) = \sum_{i=1}^n f(\boldsymbol{x}_i, \boldsymbol{q})$ over a query space $\mathcal{Q}$. A weighted subset $(S, \boldsymbol{v})$ induces the approximation $\widetilde{F}(\boldsymbol{q}) = \sum_{i\in S} v_i f(\boldsymbol{x}_i, \boldsymbol{q})$, and the goal is to ensure $\sup_{\boldsymbol{q}\in\mathcal{Q}} |F(\boldsymbol{q}) - \widetilde{F}(\boldsymbol{q})| \leq \varepsilon n$. The discrepancy approach achieves this via repeated halving: given a block of $m$ points, it finds signs $\sigma_i \in \{\pm 1\}$ such that $\sup_{\boldsymbol{q}\in\mathcal{Q}}\big|\sum_{i=1}^m \sigma_i f(\boldsymbol{x}_i, \boldsymbol{q})\big|$ is small. The points with $\sigma_i = +1$ are retained and their weights doubled. Iterating

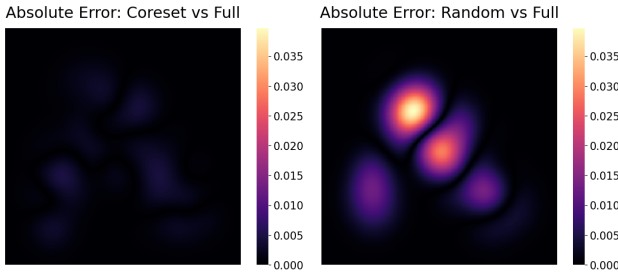

*Figure 2.* Absolute error on an imbalanced 2D GMM.

this procedure yields a compact summary while controlling the uniform approximation error.

**Discrepancy in the feature space.** Now we specialize the above view to our setting where covariate balancing is performed in the finite-dimensional feature space defined in Eq. (3). Under this representation, balancing reduces to matching feature means, naturally corresponding to controlling linear queries in the feature space. Specifically, we consider linear queries of the form $f(\boldsymbol{x}, \boldsymbol{q}) = \langle \boldsymbol{q}, \phi(\boldsymbol{x})\rangle$ over the bounded query set $\mathcal{Q}_d = \{\boldsymbol{q} \in \mathbb{R}^d : \|\boldsymbol{q}\|_2 \leq 1\}$. Then,

$$\sup_{\|\boldsymbol{q}\|_2 \leq 1}\Big|\sum_{i=1}^m \sigma_i \langle \boldsymbol{q}, \phi(\boldsymbol{x}_i)\rangle\Big| = \Big\|\sum_{i=1}^m \sigma_i\,\phi(\boldsymbol{x}_i)\Big\|_2, \quad (4)$$

where the equality follows from standard $\ell_2$–$\ell_2$ duality. Therefore, discrepancy minimization reduces to a vector balancing problem (Kulkarni et al., 2024) in $\mathbb{R}^d$.

**Greedy feature signing.** To solve the vector balancing problem in Eq. (4), we adopt a greedy signing rule that incrementally cancels the residual feature sum. Given a multiset of $2m$ feature vectors $\{\phi(\boldsymbol{x}_i)\}_{i=1}^{2m}$, we initialize $\sigma_1 = +1$ and, for $i = 2,\ldots,2m$, set

$$\sigma_i \leftarrow -\text{sign}\Big(\big\langle \sum_{j<i} \sigma_j\,\phi(\boldsymbol{x}_j),\ \phi(\boldsymbol{x}_i)\big\rangle\Big). \quad (5)$$

with the convention $\text{sign}(0) = +1$.

**Lemma 4.2** (Greedy signing residual). *Let $\{\sigma_i\}_{i=1}^{2m}$ be generated by Eq. (5). Then $\big\|\sum_{i=1}^{2m} \sigma_i \phi(\boldsymbol{x}_i)\big\|_2 \leq \sqrt{2m}$.*

*Proof.* Let $\boldsymbol{s}_0 = \boldsymbol{0}$ and $\boldsymbol{s}_i = \sum_{j\leq i} \sigma_j \phi(\boldsymbol{x}_j), \forall i \geq 1$. Then

$$\|\boldsymbol{s}_i\|_2^2 = \|\boldsymbol{s}_{i-1}\|_2^2 + \|\phi(\boldsymbol{x}_i)\|_2^2 + 2\sigma_i\langle \boldsymbol{s}_{i-1}, \phi(\boldsymbol{x}_i)\rangle.$$

By the choice of $\sigma_i$, the cross term is non-positive. Hence, $\|\boldsymbol{s}_i\|_2^2 \leq \|\boldsymbol{s}_{i-1}\|_2^2 + \|\phi(\boldsymbol{x}_i)\|_2^2$. Then, we obtain $\|\boldsymbol{s}_i\|_2^2 \leq \|\boldsymbol{s}_{i-1}\|_2^2 + 1$ since $\|\phi(\boldsymbol{x}_i)\|_2 \leq 1$. Summing over $i = [2m]$ yields $\|\boldsymbol{s}_{2m}\|_2^2 \leq 2m$, which proves the claim. $\square$

The resulting bound is deterministic and dimension-free, depending only on the number of processed points and the

uniform norm bound on the feature map. This is a key advantage of the greedy sign choice: at each step, it explicitly controls the growth of the cumulative signed feature vector, without requiring probabilistic averaging over random signs. In contrast, choosing random signs $\sigma_i \sim \text{Unif}\{\pm 1\}$ leads to a Rademacher sum (Bartlett & Mendelson, 2002), whose uniform discrepancy is typically controlled through the Rademacher complexity of the induced function class. Such complexity-based bounds may introduce additional dimension- or class-dependent factors, and can therefore be less favorable for high-dimensional random feature representations. Fig. 2 illustrates this contrast in a representative example, with experimental details provided in Appendix I.1.

**Streaming feature coresets.** With the greedy signing rule in place, we perform the halving step by retaining points with $\sigma_i = +1$ and doubling their weights. For the theoretical analysis, we consider an exact balanced halving step, where exactly $m$ out of the $2m$ points are retained, so that weight doubling preserves the total represented mass. This balanced condition can be enforced by a standard symmetrization construction: pair the $2m$ points, assign opposite signs within each pair, and apply the greedy rule to the resulting signed differences. This modification affects the discrepancy bound only up to a universal constant and therefore leaves the asymptotic error, memory, and time guarantees unchanged. For simplicity of presentation, Algorithm 1 describes the halving operation as retaining the points with $\sigma_i = +1$, while the proof uses the balanced halving formulation described above. Embedding this balanced halving operation into a standard merge-and-reduce pipeline yields the streaming coreset construction.

**Theorem 4.3** (Streaming feature coreset guarantee and complexity). *Let $(S, \boldsymbol{v})$ denote the weighted coreset maintained by Algorithm 1 after processing a stream of length $n$, using bucket size $m = \lceil 2/\varepsilon^2 \rceil$. Then the weighted feature sum represented by the coreset satisfies*

$$\left\| \sum_{i=1}^{n} \phi(\boldsymbol{x}_i) - \sum_{j:(\phi(\boldsymbol{x}_j), y_j) \in S} v_j\, \phi(\boldsymbol{x}_j) \right\|_2 \leq \varepsilon n, \quad (6)$$

*where $v_j$ is the coreset weight associated with $(\phi(\boldsymbol{x}_j), y_j) \in S$. Here, the memory cost is $\mathcal{O}(d \log n/\varepsilon^2)$, and the total computational cost is $\mathcal{O}(pdn)$ over $n$ observations.*

*Proof.* The proof is deferred to Appendix A. □

*Remark.* Our coreset construction is simple and applies to the integral feature class $\mathcal{F}$ in Assumption 4.1, rather than being limited to kernel or RKHS-based tasks.

The resulting *Feature Coreset* approximately preserves the random feature statistics required for downstream ATE estimation, while enabling a single-pass streaming implementation with only polylogarithmic overhead in the stream length. Moreover, the construction is naturally mergeable.

---

**Algorithm 1** Streaming Feature Coreset Construction

**Input:** stream $\{(\boldsymbol{x}_\tau, y_\tau)\}_{\tau \geq 1}$, bucket size $m$.
**Output:** weighted Feature Coreset $(S, \boldsymbol{v})$.

1: Initialize $\mathcal{B}_\ell \leftarrow \emptyset$ for all $\ell \geq 0$.
2: **for** $\tau = 1, 2, \ldots$ **do**
3:     Compute $\phi(\boldsymbol{x}_\tau)$ and add $((\phi(\boldsymbol{x}_\tau), y_\tau), 1)$ to $\mathcal{B}_0$.
4:     Set $\ell \leftarrow 0$.
5:     **while** $|\mathcal{B}_\ell| = 2m$ **do**      ▷ Bucket full $\Rightarrow$ compress
6:        Let $\mathcal{B}_\ell = \{((\phi(\boldsymbol{x}_i), y_i), v_i)\}_{i=1}^{2m}$.
7:        Initialize $\sigma_1 \leftarrow +1$ and $\boldsymbol{s}_1 \leftarrow \phi(\boldsymbol{x}_1)$.
8:        Set $\text{sign}(0) = +1$ with the convention.
9:        **for** $i = 2, \ldots, 2m$ **do** ▷ Greedy feature signing
10:          $\sigma_i \leftarrow -\text{sign}(\langle \boldsymbol{s}_{i-1}, \phi(\boldsymbol{x}_i) \rangle)$.
11:          $\boldsymbol{s}_i \leftarrow \boldsymbol{s}_{i-1} + \sigma_i \phi(\boldsymbol{x}_i)$.
12:        **end for**
13:        $\mathcal{C}_\ell \leftarrow \{((\phi(\boldsymbol{x}_i), y_i), 2v_i) : \sigma_i = +1\}$.
14:        $\mathcal{B}_\ell \leftarrow \emptyset$; $\mathcal{B}_{\ell+1} \leftarrow \mathcal{B}_{\ell+1} \cup \mathcal{C}_\ell$; $\ell \leftarrow \ell + 1$.
15:     **end while**
16: **end for**
17: $S \leftarrow \{(\phi(\boldsymbol{x}), y) : ((\phi(\boldsymbol{x}), y), v) \in \cup_{\ell:\mathcal{B}_\ell \neq \emptyset} \mathcal{B}_\ell\}$.
18: $\boldsymbol{v} \leftarrow \{v : ((\phi(\boldsymbol{x}), y), v) \in \cup_{\ell:\mathcal{B}_\ell \neq \emptyset} \mathcal{B}_\ell\}$.
19: **return** $(S, \boldsymbol{v})$.

---

tation with only polylogarithmic overhead in the stream length. Moreover, the construction is naturally mergeable.

**Proposition 4.4** (Mergeability). *Let $\mathcal{D}_1, \mathcal{D}_2$ be disjoint and $F_\mathcal{D}(\boldsymbol{q}) = \sum_{\boldsymbol{x} \in \mathcal{D}} f(\boldsymbol{x}, \boldsymbol{q})$. If $(S_k, \boldsymbol{v}_k)$ is an $\varepsilon_k$-coreset for $\mathcal{D}_k$, i.e., $\sup_{\boldsymbol{q} \in \mathcal{Q}} |F_{\mathcal{D}_k}(\boldsymbol{q}) - \widetilde{F}_{(S_k, \boldsymbol{v}_k)}(\boldsymbol{q})| \leq \varepsilon_k |\mathcal{D}_k|$, then the union coresets $(S_1 \cup S_2, \boldsymbol{v}_1 \cup \boldsymbol{v}_2)$ satisfies*

$$\sup_{\boldsymbol{q} \in \mathcal{Q}} |F_{\mathcal{D}_1 \cup \mathcal{D}_2}(\boldsymbol{q}) - \widetilde{F}_{(S_1 \cup S_2, \boldsymbol{v}_1 \cup \boldsymbol{v}_2)}(\boldsymbol{q})| \leq \varepsilon_1 |\mathcal{D}_1| + \varepsilon_2 |\mathcal{D}_2|.$$

*Proof.* The proof is deferred to Appendix B. □

As a consequence of Proposition 4.4, we can maintain separate Feature Coresets for the treated and control populations and merge them on demand. The merged summary remains a valid coreset for the pooled data. In the next section, we analyze the estimation error of the resulting coreset-based ATE estimator. For technical simplicity, we use the same bucket size $m$ for the treated and control groups.

### 4.2. Error Decomposition

At time $\tau$, suppose the stream prefix contains $n = n_t + n_c$ observations, with $n_t$ treated and $n_c$ control units. We maintain the corresponding weighted Feature Coresets $\mathcal{S}_t$ and $\mathcal{S}_c$. Each coreset stores the feature-outcome pairs $(\phi(\boldsymbol{x}_j), y_j)$ with coreset weights; for notational simplicity, $\mathcal{S}_t$ and $\mathcal{S}_c$ also denote their stored index sets in sums over $j$. Let $\boldsymbol{v}_t \in \mathbb{R}_+^{|\mathcal{S}_t|}$ and $\boldsymbol{v}_c \in \mathbb{R}_+^{|\mathcal{S}_c|}$ be the corresponding (unnormalized) coreset weights. We define

the normalized coreset weights $\boldsymbol{u}_t = \boldsymbol{v}_t/n_t \in \Delta^{|\mathcal{S}_t|}$, $\boldsymbol{u}_c = \boldsymbol{v}_c/n_c \in \Delta^{|\mathcal{S}_c|}$ which induce the coreset covariate measures $\mu_{t,u} := \sum_{j \in \mathcal{S}_t} u_{t,j}\delta_{\boldsymbol{x}_j}$, $\mu_{c,u} := \sum_{j \in \mathcal{S}_c} u_{c,j}\delta_{\boldsymbol{x}_j}$. Given nonnegative balancing weights $\boldsymbol{w}_t \in \mathbb{R}_+^{|\mathcal{S}_t|}$ and $\boldsymbol{w}_c \in \mathbb{R}_+^{|\mathcal{S}_c|}$, we form the final normalized weights $\boldsymbol{z}_t := \boldsymbol{u}_t \odot \boldsymbol{w}_t \in \Delta^{|\mathcal{S}_t|}$, $\boldsymbol{z}_c := \boldsymbol{u}_c \odot \boldsymbol{w}_c \in \Delta^{|\mathcal{S}_c|}$, and the corresponding weighted measures $\mu_{t,uw} := \sum_{j \in \mathcal{S}_t} u_{t,j}w_{t,j}\delta_{\boldsymbol{x}_j}$, $\mu_{c,uw} := \sum_{j \in \mathcal{S}_c} u_{c,j}w_{c,j}\delta_{\boldsymbol{x}_j}$. Here, $\boldsymbol{w}_t$ and $\boldsymbol{w}_c$ are coreset-level adjustment factors, distinct from the full-sample balancing weights $\boldsymbol{b}_t$ and $\boldsymbol{b}_c$ in Eq. (2); they need not lie in the simplex, as only the final weights $\boldsymbol{z}_t$ and $\boldsymbol{z}_c$ are normalized probability weights. In addition, we define the coreset-induced mixture measure $\mu_u := \frac{n_t}{n}\mu_{t,u} + \frac{n_c}{n}\mu_{c,u}$. Our coreset-based weighted ATE estimator is

$$\widehat{\mathrm{ATE}}_{u,w} = \langle y_1, \mu_{t,uw}\rangle_* - \langle y_0, \mu_{c,uw}\rangle_*, \qquad (7)$$

Here, the subscripts 1 and 0 serve as group indicators: $y_1$ and $y_0$ denote the observed outcomes restricted to the corresponding coresets, respectively (e.g., $\langle y_1, \mu_{t,uw}\rangle_* := \sum_{j \in \mathcal{S}_t} z_{t,j} y_j$, and $\langle y_0, \mu_{c,uw}\rangle_*$ is defined analogously).

**Decomposition.** Let $\mu := \frac{n_t}{n}\mu_t + \frac{n_c}{n}\mu_c$ denote the full-sample empirical covariate measure. By adding and subtracting $\langle r_1, \cdot\rangle_*$ and $\langle r_0, \cdot\rangle_*$, we obtain

$$\widehat{\mathrm{ATE}}_{u,w} - \mathrm{ATE} = e_{\mathrm{sam}} + e_{\mathrm{cor}} + e_{\mathrm{noi}} + e_{\mathrm{bal}}. \qquad (8)$$

Each term in Eq. (8) has a clear interpretation:

$$e_{\mathrm{sam}} := \big(\langle r_1, \mu\rangle_* - \mathbb{E}[Y_1]\big) - \big(\langle r_0, \mu\rangle_* - \mathbb{E}[Y_0]\big),$$
$$e_{\mathrm{cor}} := \langle r_1 - r_0, \mu_u - \mu\rangle_*,$$
$$e_{\mathrm{noi}} := \langle y_1 - r_1, \mu_{t,uw}\rangle_* - \langle y_0 - r_0, \mu_{c,uw}\rangle_*,$$
$$e_{\mathrm{bal}} := \langle r_1, \mu_{t,uw} - \mu_u\rangle_* - \langle r_0, \mu_{c,uw} - \mu_u\rangle_*,$$

Here, $e_{\mathrm{sam}}$ captures the finite-sample error of the full empirical mixture measure, $e_{\mathrm{cor}}$ quantifies the approximation error introduced by replacing $\mu$ with the memory-limited coreset mixture $\mu_u$, $e_{\mathrm{noi}}$ reflects the contribution of outcome noise under weighted aggregation, and $e_{\mathrm{bal}}$ measures the residual imbalance induced by the covariate balancing procedure applied to the maintained coresets.

Below we bound the four error terms separately according to their roles in the decomposition; detailed proofs are deferred to Appendices C–G. We first control the sampling error term $e_{\mathrm{sam}}$, which arises from replacing the population covariate distribution with its full-sample empirical counterpart.

**Lemma 4.5** (Finite-sample error). *For any $\delta \in (0,1)$, with probability at least $1 - \delta$, $|e_{\mathrm{sam}}| \leq 2C\sqrt{2\log(4/\delta)/n}$.*

Lemma 4.5 shows that the full-sample empirical covariate measure approximates the population-level target at the standard rate $\mathcal{O}(n^{-1/2})$. We next turn to the coreset-induced approximation error $e_{\mathrm{cor}}$. To control $e_{\mathrm{cor}}$, we introduce a discrepancy metric for the integral feature class.

**Definition 4.6** ($\mathcal{F}$-discrepancy). For two measures $\nu$ and $\nu'$ on $\mathcal{X}$, define the $\mathcal{F}$-discrepancy between $\nu$ and $\nu'$ as

$$d_{\mathcal{F}}(\nu, \nu') := \sup_{f \in \mathcal{F}} \left| \int_{\mathcal{X}} f(\boldsymbol{x}) \, d(\nu - \nu')(\boldsymbol{x}) \right|. \qquad (9)$$

The discrepancy $d_{\mathcal{F}}$ in Definition 4.6 is defined with respect to the function class $\mathcal{F}$ in Assumption 4.1. For a general integral feature class, obtaining a finite-dimensional random-feature approximation to $d_{\mathcal{F}}$ may require uniform concentration over $\mathcal{F}$ together with additional complexity control, such as metric entropy or bracketing arguments. To keep the analysis transparent, we specialize to the case where $\mathcal{F}$ is the radius-$C$ ball of an RKHS associated with the kernel induced by $\psi$. Under this specialization, the random feature map in Eq. (3) provides a tractable approximation to the RKHS discrepancy and yields a $d^{-1/4}$ Monte Carlo rate, as stated in the following lemma.

**Lemma 4.7** ($\mathcal{F}$-discrepancy upper bound). *Define the mean feature embedding $\Phi_d(\nu) := \int_{\mathcal{X}} \phi(\boldsymbol{x}) \, d\nu(\boldsymbol{x})$. For any $\delta \in (0,1)$, with probability at least $1 - \delta$,*

$$d_{\mathcal{F}}(\nu, \nu') \leq C\big\|\Phi_d(\nu) - \Phi_d(\nu')\big\|_2 + C\left(\frac{8\log(2/\delta)}{d}\right)^{1/4}. \qquad (10)$$

*Remark.* The RKHS-ball specialization is used only to obtain a clean finite-dimensional discrepancy bound, not to restrict the practical implementation to RKHS-based outcome models. If the true outcome regression function lies outside $\mathcal{F}$, the analysis can be viewed relative to its best approximation in $\mathcal{F}$, adding an approximation residual while leaving the coreset and balancing errors controlled. Thus, under outcome shifts, the error degrades additively rather than failing abruptly. Our experiments further support this behavior by testing the method under complex nonlinear outcome shifts beyond simple RKHS-ball specifications.

Moreover, by the mergeability property of coresets in Proposition 4.4, together with the streaming feature coreset guarantee in Theorem 4.3, the maintained coreset mixture $\mu_u$ satisfies $\big\|\Phi_d(\mu_u) - \Phi_d(\mu)\big\|_2 \leq \varepsilon$. Since $|e_{\mathrm{cor}}| \leq 2\,d_{\mathcal{F}}(\mu_u, \mu)$, we immediately obtain the following bound.

**Corollary 4.8** (Coreset-induced error bound). *For any $\delta \in (0,1)$, with probability at least $1 - \delta$,*

$$|e_{\mathrm{cor}}| \leq 2\,d_{\mathcal{F}}(\mu_u, \mu) \leq 2C\varepsilon + 2C(8\log(2/\delta)/d)^{1/4}. \qquad (11)$$

We next control the noise-induced error $e_{\mathrm{noi}}$ under a standard sub-Gaussian noise condition (Wainwright, 2019).

**Assumption 4.9** (Outcome noise). For each sampled unit and treatment level $a \in \{0, 1\}$, define $\eta_{a,i} := y_{a,i} - r_a(\boldsymbol{x}_i)$. Conditional on the covariates, these outcome residuals are independent, mean-zero, and $\sigma^2$-sub-Gaussian.

Then we have the following bound for $e_{\text{noi}}$.

**Lemma 4.10** (Outcome noise error). *Under Assumption 4.9, for any $\delta \in (0, 1)$, with probability at least $1 - \delta$,*

$$|e_{\text{noi}}| \leq \sigma \sqrt{2 \log(4/\delta)} \left( \sqrt{\sum_{j \in \mathcal{S}_t} z_{t,j}^2} + \sqrt{\sum_{j \in \mathcal{S}_c} z_{c,j}^2} \right).$$
(12)

Lemma 4.10 shows that the noise-induced error depends on the $\ell_2$ norms of the final normalized weights. Highly concentrated weights, such as $z_{t,j^*} = 1$ for some $j^*$, can therefore lead to substantial variance amplification. This motivates regularizing the balancing weights to avoid weight collapse, as discussed further in Sec. 4.4.

The remaining term $e_{\text{bal}}$ captures the residual covariate imbalance induced by the balancing procedure. Unlike $e_{\text{sam}}$, $e_{\text{cor}}$, and $e_{\text{noi}}$, this term can be directly controlled by appropriately choosing the balancing weights $(\boldsymbol{w}_t, \boldsymbol{w}_c)$ on the maintained feature coresets. The following lemma relates $e_{\text{bal}}$ to discrepancies between feature mean embeddings.

**Lemma 4.11** (Balancing error bound). *For any $\delta \in (0, 1)$, with probability at least $1 - \delta$,*

$$|e_{\text{bal}}| \leq C \|\Phi_d(\mu_{t,uw}) - \Phi_d(\mu_u)\|_2$$
$$+ C \|\Phi_d(\mu_{c,uw}) - \Phi_d(\mu_u)\|_2 + 2C \left( \frac{8 \log(4/\delta)}{d} \right)^{1/4}.$$
(13)

Lemma 4.11 shows that, up to a random-feature approximation term of order $\mathcal{O}(d^{-1/4})$, controlling the balancing error reduces to matching the feature mean embeddings of the weighted treated and control measures to that of the coreset-induced mixture distribution.

### 4.3. Covariate Balancing on Feature Coresets

As shown above, controlling the balancing error $e_{\text{bal}}$ reduces to matching the feature mean embeddings of the weighted treated and control measures to that of the coreset-induced mixture. We therefore define the feature balancing objective as

$$\mathcal{L}_\Phi := \left\| \Phi_d(\mu_{t,uw}) - \Phi_d(\mu_u) \right\|_2^2 + \left\| \Phi_d(\mu_{c,uw}) - \Phi_d(\mu_u) \right\|_2^2.$$
(14)

**Closed-form quadratic objective.** We rewrite Eq. (14) in matrix form. Let $U := \mathcal{S}_t \cup \mathcal{S}_c$ be the maintained buffer, ordered with treated coreset points first and control coreset points second, and define $\boldsymbol{\Phi}_U := [\phi(\boldsymbol{x}_j)]_{j \in U} \in \mathbb{R}^{d \times |U|}$. Recall that $\mu_u = \frac{n_t}{n} \mu_{t,u} + \frac{n_c}{n} \mu_{c,u}$. Under the ordering of $U$, let $\boldsymbol{\beta} \in \mathbb{R}^{|U|}$ denote the mixture-weight vector with treated block $\frac{n_t}{n} \boldsymbol{u}_t$ and control block $\frac{n_c}{n} \boldsymbol{u}_c$. Define the embedded treated and control weight vectors on $U$ by $\boldsymbol{\alpha}_t :=$

$\mathbf{A}_t \boldsymbol{w}_t$ and $\boldsymbol{\alpha}_c := \mathbf{A}_c \boldsymbol{w}_c$, where $\mathbf{A}_t := \left[ \text{diag}(\boldsymbol{u}_t); \mathbf{0} \right]^\top \in \mathbb{R}^{|U| \times |\mathcal{S}_t|}$, $\mathbf{A}_c := \left[ \mathbf{0}; \text{diag}(\boldsymbol{u}_c) \right]^\top \in \mathbb{R}^{|U| \times |\mathcal{S}_c|}$. Then $\Phi_d(\mu_{t,uw}) = \boldsymbol{\Phi}_U \boldsymbol{\alpha}_t$, $\Phi_d(\mu_{c,uw}) = \boldsymbol{\Phi}_U \boldsymbol{\alpha}_c$, and $\Phi_d(\mu_u) = \boldsymbol{\Phi}_U \boldsymbol{\beta}$. Thus, the balancing objective becomes

$$\mathcal{L}_\Phi = \left\| \boldsymbol{\Phi}_U (\boldsymbol{\alpha}_t - \boldsymbol{\beta}) \right\|_2^2 + \left\| \boldsymbol{\Phi}_U (\boldsymbol{\alpha}_c - \boldsymbol{\beta}) \right\|_2^2.$$
(15)

Expanding Eq. (15) and dropping constants independent of $(\boldsymbol{w}_t, \boldsymbol{w}_c)$ gives the quadratic objective

$$\mathcal{L} := \boldsymbol{w}_t^\top \mathbf{Q}_t \boldsymbol{w}_t - 2\boldsymbol{q}_t^\top \boldsymbol{w}_t + \boldsymbol{w}_c^\top \mathbf{Q}_c \boldsymbol{w}_c - 2\boldsymbol{q}_c^\top \boldsymbol{w}_c, \quad (16)$$

where $\mathbf{Q}_t := \mathbf{A}_t^\top \boldsymbol{\Phi}_U^\top \boldsymbol{\Phi}_U \mathbf{A}_t$, $\mathbf{Q}_c := \mathbf{A}_c^\top \boldsymbol{\Phi}_U^\top \boldsymbol{\Phi}_U \mathbf{A}_c$, and $\boldsymbol{q}_t := \mathbf{A}_t^\top \boldsymbol{\Phi}_U^\top \boldsymbol{\Phi}_U \boldsymbol{\beta}$, $\boldsymbol{q}_c := \mathbf{A}_c^\top \boldsymbol{\Phi}_U^\top \boldsymbol{\Phi}_U \boldsymbol{\beta}$.

**Proposition 4.12.** *The objective $\mathcal{L}$ is convex in $(\boldsymbol{w}_t, \boldsymbol{w}_c)$.*

*Proof.* Since $\mathbf{Q}_t = \mathbf{A}_t^\top (\boldsymbol{\Phi}_U^\top \boldsymbol{\Phi}_U) \mathbf{A}_t \succeq \mathbf{0}$ and $\mathbf{Q}_c = \mathbf{A}_c^\top (\boldsymbol{\Phi}_U^\top \boldsymbol{\Phi}_U) \mathbf{A}_c \succeq \mathbf{0}$, the quadratic terms are convex. Hence, $\mathcal{L}$ is a sum of convex quadratic terms and linear terms in $(\boldsymbol{w}_t, \boldsymbol{w}_c)$, and is therefore convex. $\square$

**Online optimization under simplex constraints.** The final group-specific weights lie on probability simplices, with $\boldsymbol{z}_t = \boldsymbol{u}_t \odot \boldsymbol{w}_t \in \Delta^{|\mathcal{S}_t|}$ and $\boldsymbol{z}_c = \boldsymbol{u}_c \odot \boldsymbol{w}_c \in \Delta^{|\mathcal{S}_c|}$. Since the objective $\mathcal{L}$ is convex (Proposition 4.12) and the feasible sets are compact simplices, the problem admits standard online convex optimization algorithms. Let $T$ denote the number of online updates (iterations). In particular, under bounded gradients, mirror descent with the negative-entropy mirror map, equivalently exponentiated-gradient updates, enjoys the standard $\mathcal{O}(\sqrt{T})$ regret bound over the simplex (Shalev-Shwartz, 2025). For a fixed convex balancing objective, the usual online-to-batch conversion then yields $\mathcal{O}(1/\sqrt{T})$ suboptimality for the averaged iterate. We use this observation as a theoretical justification for applying simplex-constrained optimization to the balancing step.

**Practical optimization.** Although mirror descent provides clean theoretical guarantees, its empirical performance may be sensitive to step-size selection and other hyperparameters, especially under online updates. Motivated by this practical consideration, we use an unconstrained softmax parameterization, which enforces the simplex constraints by construction and can be optimized conveniently with adaptive first-order methods (Kingma & Ba, 2014).

Specifically, for each group $a \in \{t, c\}$, we introduce logits $\boldsymbol{g}_a \in \mathbb{R}^{|\mathcal{S}_a|}$ and set $\boldsymbol{z}_a(\boldsymbol{g}_a) := \text{softmax}(\boldsymbol{g}_a) \in \Delta^{|\mathcal{S}_a|}$. We then optimize $\mathcal{L}$ with respect to $\boldsymbol{g}_t$ and $\boldsymbol{g}_c$ using Adam, warm-started from the solution at the previous time step to encourage stable online updates. Empirically, this softmax-Adam implementation often provides faster and more stable minimization of the balancing objective than mirror descent; additional comparisons are provided in Appendix J.

## 4.4. Overall Algorithm

We now present the complete method, termed **FCore** (*Feature Coresets for streaming covariate balancing*). FCore combines discrepancy-preserving feature coresets with online covariate balancing to produce anytime ATE estimates under a fixed memory budget.

---

**Algorithm 2** FCore: Streaming ATE Estimation

---

**Input:** data stream $\{(\boldsymbol{x}_\tau, a_\tau, y_\tau)\}_{\tau \geq 1}$, bucket size $m$.

**Output:** Prefix ATE estimates $\{\widehat{\text{ATE}}_\tau\}_{\tau \geq 1}$.

1: Initialize treated / control coresets $\mathcal{S}_t \leftarrow \emptyset$, $\mathcal{S}_c \leftarrow \emptyset$.
2: **for** $\tau = 1, 2, \ldots$ **do**
3:      Observe $(\boldsymbol{x}_\tau, a_\tau, y_\tau)$ and compute $\phi(\boldsymbol{x}_\tau)$.
4:      **if** $a_\tau = 1$ **then**
5:          Insert $(\phi(\boldsymbol{x}_\tau), y_\tau)$ into $\mathcal{S}_t$ via Algorithm 1.
6:      **else**
7:          Insert $(\phi(\boldsymbol{x}_\tau), y_\tau)$ into $\mathcal{S}_c$ via Algorithm 1.
8:      **end if**
9:      Covariate balancing on Feature Coresets $\mathcal{S}_t \cup \mathcal{S}_c$.
10:     Compute $\widehat{\text{ATE}}_\tau$ using Eq. (7).
11: **end for**

---

Algorithm 2 summarizes the procedure. At each time step, FCore updates group-specific feature coresets (Lines 4-8), optimizes balancing weights over the maintained coresets (Line 9), and produces a estimate of prefix ATE (Line 10).

**Complexity guarantee.** Let $n$ be the stream length and set bucket size $m = \lceil 2/\varepsilon^2 \rceil$. FCore maintains feature coresets of total size $\mathcal{O}(\log n/\varepsilon^2)$, resulting in $\mathcal{O}(d \log n/\varepsilon^2)$ memory usage. Processing each sample requires $\mathcal{O}(pd)$ time to compute random features. Covariate balancing is performed on the maintained coresets and incurs $\mathcal{O}(d \log n/\varepsilon^2)$ time per objective update. Consequently, excluding balancing iterations, FCore requires $\mathcal{O}(pdn)$ total feature-computation time, while its memory usage and per-step balancing cost scale only polylogarithmically with $n$.

**Choice of feature map $\phi(\boldsymbol{x})$.** The feature map $\phi(\boldsymbol{x})$ can be instantiated using a broad class of randomized constructions, including random Fourier features, random neural features (e.g., single-layer networks with random weights), and other randomized basis expansions. In practice, we adopt random Fourier features (RFFs) for their computational efficiency and simplicity (Rahimi & Recht, 2007; Sutherland & Schneider, 2015). Specifically, we draw $d/2$ frequencies $\boldsymbol{\omega}_r \overset{\text{i.i.d.}}{\sim} \mathcal{N}(\boldsymbol{0}, \gamma \mathbf{I}_p)$ and define the feature map $\phi(\boldsymbol{x}) = \sqrt{\frac{2}{d}} \big[ \cos(\boldsymbol{\omega}_r^\top \boldsymbol{x}), \sin(\boldsymbol{\omega}_r^\top \boldsymbol{x}) \big]_{r=1}^{d/2}$, where $[\cdot]_{r=1}^{d/2}$ denotes concatenation. The convergence analysis in this work is developed under the RKHS-ball setting, under which the RFF construction employed here is theoretically justified.

**Main theoretical result.** We now summarize the main theoretical guarantee of the proposed framework. Combining Lemma 4.5, Corollary 4.8, Lemma 4.10, and Lemma 4.11, we obtain that, with high probability,

$$
\begin{aligned}
\left| \widehat{\text{ATE}}_{u,w} - \text{ATE} \right| &= \mathcal{O}\left( \varepsilon + \frac{1}{d^{1/4}} + \frac{1}{\sqrt{n}} \right) \\
&+ \mathcal{O}\left( \sqrt{\mathcal{L}_\Phi} \right) + \mathcal{O}\left( \sqrt{\sum_{j \in \mathcal{S}_t} z_{t,j}^2} + \sqrt{\sum_{j \in \mathcal{S}_c} z_{c,j}^2} \right),
\end{aligned}
\tag{17}
$$

The bound indicates that the ATE error is governed by two controllable quantities: the balancing loss $\mathcal{L}_\Phi$, which captures residual covariate imbalance, and the concentration of the normalized weights, which controls the variance. Reducing $\mathcal{L}_\Phi$ shrinks the bias term, while preventing weight collapse limits the variance. Although entropy regularization (Yan et al., 2024; Hainmueller, 2012) provides a principled remedy, in practice we directly monitor weight concentration to detect and mitigate collapse.

## 5. Experiments

We compare **FCore** with several online causal inference baselines (Luo et al., 2024): G-computation (**G-comp**), which estimates the average treatment effect via outcome regression; **IPTW**, which uses inverse propensity score weighting; and **AIPTW**, which combines outcome regression and inverse probability weighting to form a doubly robust estimator. Due to space limitations, detailed experimental setup, runtime results, parameter sensitivity analysis, and comparisons with additional baselines are provided in Appendices I and J. Our code is available at https://github.com/jinchenghou123/FCore.

For evaluating the methods, we adopt the mean absolute errors (MAE) $|\widehat{\text{ATE}} - \text{ATE}|$ as the performance metric. We carry out each experiments 100 times and report the mean and standard deviation of the corresponding results.

**Performance on synthetic datasets.** We evaluate the proposed method on two synthetic streaming benchmarks designed to capture different types of distributional shift. The *Heterogeneous outcome shift* dataset introduces non-stationarity in the outcome-generating process through increasing functional complexity and rotating heterogeneous treatment effects. The *Kang–Schafer treatment shift* dataset follows a classical latent confounding construction (Kang & Schafer, 2007) and induces a pure shift in the treatment assignment mechanism by varying the propensity intercept across batches while keeping the outcome model fixed. In both settings, data arrive sequentially in $B = 5$ batches, and we report results at the final prefix after observing all batches. Additional details on data generation and prefix-wise evaluation are provided in Appendix I.2.

Fig. 3 shows MAE results at the final batch for the *Heterogeneous outcome shift* and *Kang–Schafer treatment shift* datasets, respectively. Under heterogeneous outcome shift, model-based baselines (G-comp, IPTW, and AIPTW) exhibit substantial estimation error due to outcome model misspecification, whereas FCore achieves consistently lower MAE, demonstrating robustness to evolving outcome-generating mechanisms. Under Kang–Schafer treatment shift, propensity-score-based methods suffer from large errors and high variance as treatment prevalence changes over time, while FCore maintains near-zero MAE across different coreset retention ratios. On both datasets, FCore remains stable even under aggressive coreset compression, highlighting its ability to handle distributional shifts while operating under strict memory constraints.

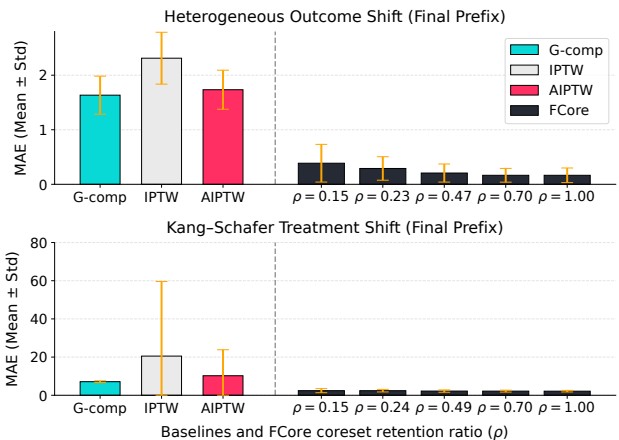

*Figure 3.* Performance comparison on synthetic data. Top: Heterogeneous outcome shift. Below: Kang–Schafer treatment shift.

**Performance on real datasets.** We evaluate the proposed method on two real-world datasets: the Infant Health and Development Program (IHDP) (Dorie, 2016) and a COVID-19 vaccine safety monitoring dataset[1]. The IHDP dataset studies the causal effect of specialist home visits on infants' future cognitive scores. Following the standard Setting A protocol (Dorie, 2016), we extend it to a streaming setting by treating the 10 independent replications as sequential batches. For the COVID-19 dataset, we estimate the comparative risk of headache and fatigue between the Pfizer and Moderna vaccines using data from 2021, and process the data as a stream of 12 monthly batches to update effect estimates over time. Due to space limit, additional details on the datasets are provided in Appendix I.3.

On the IHDP dataset (Fig. 4), FCore consistently achieves lower MAE than G-comp, IPTW, and AIPTW across all coreset retention ratios, and remains accurate even under aggressive compression ($\rho = 0.16$). On the COVID-19 dataset (Fig. 5), FCore recovers a negative ATE of approxi-

[1] https://vaers.hhs.gov/data.html

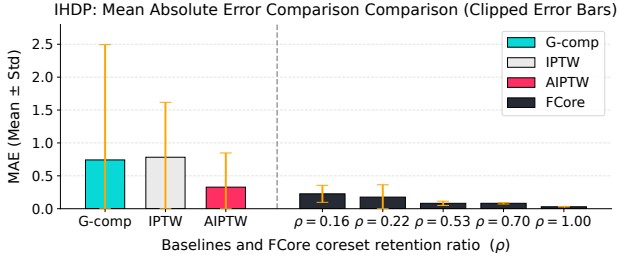

*Figure 4.* Performance comparison on the IHDP dataset.

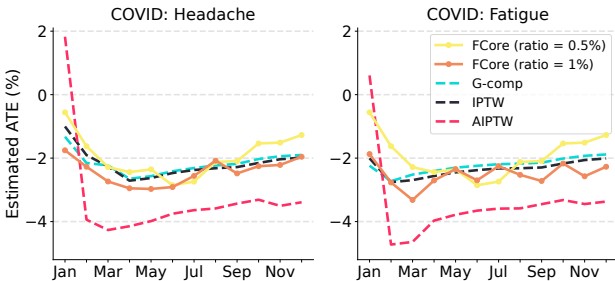

*Figure 5.* Performance comparison on the COVID-19 dataset. Left: headache. Right: fatigue. The ratio means the retention ratio.

mately $-2\%$ for both headache and fatigue while using only $0.5\%$–$1\%$ of the data, consistent with previously reported findings (Luo et al., 2024). These results demonstrate the effectiveness of FCore for streaming causal inference under severe memory constraints.

## 6. Conclusion

This paper proposes a model-agnostic method for causal inference in streaming setting with memory constraint and distributional shift. With discrepancy-based feature coresets, the method enables covariate balancing over a rich nonparametric function space with linear-time updates and bounded memory, while avoiding parametric nuisance models. Theoretical guarantees and empirical results on synthetic and real-world datasets demonstrate its robust and accurate performance. Future work includes exploring task-informed feature constructions to further improve performance.

## Acknowledgements

Yixin Ren, Chenghou Jin, Yewei Xia, and Shuigeng Zhou were supported by National Natural Scinece Foundation of China (NSFC) (No. 62372116), Jihong Guan was supported by NSFC (No. 62372326), Yixin Ren was also supported by Tencent Rhino-Bird Elite Talent Program. The computations in this research were partially performed using the CFFF platform of Fudan University. The authors would like to thank the anonymous reviewers for their valuable advice.

## Impact Statement

This paper presents work whose goal is to advance the field of Machine Learning and Causality. There are many potential societal consequences of our work, none which we feel must be specifically highlighted here.

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

# Appendix Organization

## A. Proof of Theorem 4.3

For convenience, we restate Theorem 4.3. All notation introduced in this part is local and used only for the proof.

**Theorem A.1** (Streaming Feature Coreset guarantee and complexity). *Let $(S, \boldsymbol{v})$ denote the weighted coreset maintained by Algorithm 1 after processing a stream of length $n$, using bucket size $m = \lceil 2/\varepsilon^2 \rceil$. Then the weighted feature sum represented by the coreset satisfies*

$$\Big\| \sum_{i=1}^{n} \phi(\boldsymbol{x}_i) - \sum_{j:(\phi(\boldsymbol{x}_j), y_j) \in S} v_j \, \phi(\boldsymbol{x}_j) \Big\|_2 \ \leq \ \varepsilon n, \tag{18}$$

*where $v_j$ is the coreset weight associated with $(\phi(\boldsymbol{x}_j), y_j) \in S$. Here, the memory cost is $\mathcal{O}(d \log n/\varepsilon^2)$, and the total computational cost is $\mathcal{O}(pdn)$ over $n$ observations.*

*Proof.* We prove the approximation guarantee in Eq. (18) by tracking the error introduced by each *compression* step in the merge-and-reduce procedure and showing that the accumulated error remains bounded by $\varepsilon n$. The argument proceeds in two steps. First, we bound the error contributed by a single final bucket by summing the compression errors along its merge-and-reduce history. Second, we aggregate these bucket-level errors over all nonempty levels, using the fact that higher-level buckets represent disjoint subsets of the original stream and carry geometrically increasing weights. Together, these steps yield the desired global feature-sum approximation.

**Setup and merge-and-reduce invariant.** Algorithm 1 maintains buckets $\{\mathcal{B}_\ell\}_{\ell \geq 0}$. Under the balanced halving convention used in the analysis and described in Sec. 4.1, each compression step assigns signs $\sigma_i \in \{\pm 1\}$ with exactly $m$ positive and $m$ negative signs to the $2m$ items. The algorithm keeps the $m$ items with $\sigma_i = +1$ and doubles their weights from $2^\ell$ to $2^{\ell+1}$ before inserting them into level $\ell + 1$. Thus, each compression preserves the total represented mass. As in standard merge-and-reduce schemes, after processing $n$ points, there is at most one nonempty bucket at each level.

**Accumulating error over the whole stream.** Let $\boldsymbol{\Phi}_{\text{full}} := \sum_{i=1}^{n} \phi(\boldsymbol{x}_i)$ and $\boldsymbol{\Phi}_{\text{core}} := \sum_{j:(\phi(\boldsymbol{x}_j), y_j) \in S} v_j \, \phi(\boldsymbol{x}_j)$ denote the full feature sum and the weighted feature sum represented by the final coreset, respectively, where $v_j$ is the corresponding coreset weight associated with $(\phi(\boldsymbol{x}_j), y_j) \in S$.

We analyze the merge-and-reduce structure at the end of the stream. By construction, for each level $\ell \geq 0$, there is at most one nonempty bucket $\mathcal{B}_\ell$, and every item stored in $\mathcal{B}_\ell$ carries weight $2^\ell$. Whenever a level-$\ell$ compression occurs, $2m$ items with weight $2^\ell$ are replaced by $m$ items with weight $2^{\ell+1}$. Fix a level $\ell$ such that $\mathcal{B}_\ell$ is nonempty at the end of the stream. Let $\{\phi(\boldsymbol{x}_{\ell,i})\}_{i=1}^{t_\ell}$ denote the unweighted feature vectors stored in $\mathcal{B}_\ell$, where $t_\ell \leq 2m$, each carrying weight $2^\ell$. Define the weighted feature contribution of this final bucket as $\boldsymbol{b}_\ell := 2^\ell \sum_{i=1}^{t_\ell} \phi(\boldsymbol{x}_{\ell,i})$. Then the final coreset feature sum decomposes as $\boldsymbol{\Phi}_{\text{core}} = \sum_{\ell \geq 0} \boldsymbol{b}_\ell$, where the sum has only finitely many nonzero terms.

Now consider the original points represented by the final bucket $\mathcal{B}_\ell$. Let $\boldsymbol{a}_\ell$ denote their true feature sum, where each original point is counted once with weight 1. The difference between $\boldsymbol{b}_\ell$ and $\boldsymbol{a}_\ell$ is the accumulated effect of the compression errors along the merge-and-reduce paths that produce the items in $\mathcal{B}_\ell$. Along any such path, there is at most one compression

at each intermediate level. At level $k$, the compression involves items of weight $2^k$, so Lemma 4.2 bounds the corresponding weighted error by $2^k\sqrt{2m}$. For $\ell \geq 1$, summing over levels $k = 0, \ldots, \ell - 1$ gives

$$\|\boldsymbol{b}_\ell - \boldsymbol{a}_\ell\|_2 \leq \sum_{k=0}^{\ell-1} 2^k \sqrt{2m} = (2^\ell - 1)\sqrt{2m} < 2^\ell \sqrt{2m}. \tag{19}$$

For $\ell = 0$, no compression has occurred, so $\|\boldsymbol{b}_0 - \boldsymbol{a}_0\|_2 = 0$, consistent with the bound.

Finally, the original stream is partitioned by the final buckets across levels, so $\boldsymbol{\Phi}_{\text{full}} = \sum_{\ell \geq 0} \boldsymbol{a}_\ell$ and $\boldsymbol{\Phi}_{\text{core}} = \sum_{\ell \geq 0} \boldsymbol{b}_\ell$. Therefore, by the triangle inequality and Eq. (19),

$$\|\boldsymbol{\Phi}_{\text{full}} - \boldsymbol{\Phi}_{\text{core}}\|_2 = \left\|\sum_{\ell \geq 0}(\boldsymbol{a}_\ell - \boldsymbol{b}_\ell)\right\|_2 = \left\|\sum_{\ell \geq 1}(\boldsymbol{a}_\ell - \boldsymbol{b}_\ell)\right\|_2 \leq \sum_{\ell \geq 1}\|\boldsymbol{a}_\ell - \boldsymbol{b}_\ell\|_2 \leq \sum_{\ell \geq 1} 2^\ell \sqrt{2m} \cdot \mathbf{1}\{\mathcal{B}_\ell \neq \emptyset\}. \tag{20}$$

The level-0 term is omitted because it incurs no compression error. For every $\ell \geq 1$, if $\mathcal{B}_\ell$ is nonempty at the end of the stream, then it has undergone at least one halving step and therefore contains at least $m$ items, each representing $2^\ell$ original samples. Since the final buckets summarize disjoint subsets of the stream, we have $\sum_{\ell \geq 1} m \cdot 2^\ell \cdot \mathbf{1}\{\mathcal{B}_\ell \neq \emptyset\} \leq n$. This implies $\sum_{\ell \geq 1} 2^\ell \cdot \mathbf{1}\{\mathcal{B}_\ell \neq \emptyset\} \leq \frac{n}{m}$. Substituting into Eq. (20) yields

$$\|\boldsymbol{\Phi}_{\text{full}} - \boldsymbol{\Phi}_{\text{core}}\|_2 \leq \sqrt{2m} \cdot \frac{n}{m} = \frac{\sqrt{2}\,n}{\sqrt{m}}. \tag{21}$$

Choosing $m = \lceil 2/\varepsilon^2 \rceil$ gives $\frac{\sqrt{2}}{\sqrt{m}} \leq \varepsilon$, and therefore $\|\boldsymbol{\Phi}_{\text{full}} - \boldsymbol{\Phi}_{\text{core}}\|_2 \leq \varepsilon n$.

**Memory and time complexity.** The merge-and-reduce pipeline maintains at most one bucket per level, each storing at most $2m$ items. The number of levels is $\mathcal{O}(\log n)$, yielding coreset size $\mathcal{O}(m \log n)$ and memory

$$\mathcal{O}(d\, m \log n) = \mathcal{O}\big(d \log n/\varepsilon^2\big).$$

For runtime, computing the random feature map $\phi(\boldsymbol{x}_i) \in \mathbb{R}^d$ costs $\mathcal{O}(pd)$ per sample, for a total of $\mathcal{O}(pdn)$. Each compression processes feature vectors in time linear in their number and in the feature dimension. Across all levels, the total number of items processed by compression is bounded by a geometric series, and is therefore $\mathcal{O}\big(n + n/2 + n/4 + \cdots\big) = \mathcal{O}(n)$. Thus, the total time spent on coreset maintenance is $\mathcal{O}(dn)$. Combining feature computation and coreset maintenance yields an overall time complexity of $\mathcal{O}(pdn)$. $\square$

## B. Proof of Proposition 4.4

For convenience, we restate Proposition 4.4. All notation introduced in this part is local and used only for the proof.

**Proposition B.1** (Mergeability). *Let $\mathcal{D}_1, \mathcal{D}_2$ be disjoint and $F_{\mathcal{D}}(\boldsymbol{q}) = \sum_{\boldsymbol{x} \in \mathcal{D}} f(\boldsymbol{x}, \boldsymbol{q})$. If $(S_k, \boldsymbol{v}_k)$ is an $\varepsilon_k$-coreset for $\mathcal{D}_k$, i.e., $\sup_{\boldsymbol{q} \in \mathcal{Q}} |F_{\mathcal{D}_k}(\boldsymbol{q}) - \widetilde{F}_{(S_k, \boldsymbol{v}_k)}(\boldsymbol{q})| \leq \varepsilon_k |\mathcal{D}_k|$, then the union coresets $(S_1 \cup S_2, \boldsymbol{v}_1 \cup \boldsymbol{v}_2)$ satisfies*

$$\sup_{\boldsymbol{q} \in \mathcal{Q}} |F_{\mathcal{D}_1 \cup \mathcal{D}_2}(\boldsymbol{q}) - \widetilde{F}_{(S_1 \cup S_2, \boldsymbol{v}_1 \cup \boldsymbol{v}_2)}(\boldsymbol{q})| \leq \varepsilon_1 |\mathcal{D}_1| + \varepsilon_2 |\mathcal{D}_2|.$$

*Proof.* Since $\mathcal{D}_1$ and $\mathcal{D}_2$ are disjoint, $F_{\mathcal{D}_1 \cup \mathcal{D}_2}(\boldsymbol{q}) = F_{\mathcal{D}_1}(\boldsymbol{q}) + F_{\mathcal{D}_2}(\boldsymbol{q})$ for all $\boldsymbol{q} \in \mathcal{Q}$. By the definition of the union coreset $(S, \boldsymbol{v})$, $\widetilde{F}_{(S, \boldsymbol{v})}(\boldsymbol{q}) = \widetilde{F}_{(S_1, \boldsymbol{v}_1)}(\boldsymbol{q}) + \widetilde{F}_{(S_2, \boldsymbol{v}_2)}(\boldsymbol{q})$. Therefore, for any $\boldsymbol{q} \in \mathcal{Q}$,

$$\left|F_{\mathcal{D}_1 \cup \mathcal{D}_2}(\boldsymbol{q}) - \widetilde{F}_{(S, \boldsymbol{v})}(\boldsymbol{q})\right| = \left|\big(F_{\mathcal{D}_1}(\boldsymbol{q}) - \widetilde{F}_{(S_1, \boldsymbol{v}_1)}(\boldsymbol{q})\big) + \big(F_{\mathcal{D}_2}(\boldsymbol{q}) - \widetilde{F}_{(S_2, \boldsymbol{v}_2)}(\boldsymbol{q})\big)\right|$$
$$\leq \left|F_{\mathcal{D}_1}(\boldsymbol{q}) - \widetilde{F}_{(S_1, \boldsymbol{v}_1)}(\boldsymbol{q})\right| + \left|F_{\mathcal{D}_2}(\boldsymbol{q}) - \widetilde{F}_{(S_2, \boldsymbol{v}_2)}(\boldsymbol{q})\right|.$$

Taking the supremum over $\boldsymbol{q} \in \mathcal{Q}$ and applying the two coreset guarantees gives

$$\sup_{\boldsymbol{q} \in \mathcal{Q}} \left|F_{\mathcal{D}_1 \cup \mathcal{D}_2}(\boldsymbol{q}) - \widetilde{F}_{(S, \boldsymbol{v})}(\boldsymbol{q})\right| \leq \varepsilon_1 |\mathcal{D}_1| + \varepsilon_2 |\mathcal{D}_2|.$$

This establishes the desired mergeability guarantee and completes the proof. $\square$

# C. Proof of Lemma 4.5

For convenience, we restate Lemma 4.5. All notation introduced in this part is local and used only for the proof.

**Lemma C.1** (Finite-sample error)**.** *For any $\delta \in (0, 1)$, with probability at least $1 - \delta$,*

$$|e_{\text{sam}}| = \left|\left(\langle r_1, \mu \rangle_* - \mathbb{E}[Y_1]\right) - \left(\langle r_0, \mu \rangle_* - \mathbb{E}[Y_0]\right)\right| \leq 2C\sqrt{\frac{2\log(4/\delta)}{n}}. \tag{22}$$

*Proof.* We first show that Assumption 4.1 implies the uniform boundedness of the outcome regression functions.

**Boundedness of $r_a(\boldsymbol{x})$.** Fix $a \in \{0, 1\}$ and $\boldsymbol{x} \in \mathcal{X}$. Since $r_a \in \mathcal{F}$, there exists a measurable function $o_a(\boldsymbol{h})$ such that

$$r_a(\boldsymbol{x}) = \int_\Omega o_a(\boldsymbol{h}) \, \psi(\boldsymbol{x}; \boldsymbol{h}) \, \mathrm{d}\boldsymbol{h}, \quad |o_a(\boldsymbol{h})| \leq Cp(\boldsymbol{h}) \text{ for all } \boldsymbol{h} \in \Omega. \tag{23}$$

Using the triangle inequality and the bound on $o_a$, we obtain

$$|r_a(\boldsymbol{x})| \leq \int_\Omega |o_a(\boldsymbol{h})| \, |\psi(\boldsymbol{x}; \boldsymbol{h})| \, \mathrm{d}\boldsymbol{h} \leq \int_\Omega Cp(\boldsymbol{h}) \, |\psi(\boldsymbol{x}; \boldsymbol{h})| \, \mathrm{d}\boldsymbol{h}. \tag{24}$$

By Assumption 4.1, $|\psi(\boldsymbol{x}; \boldsymbol{h})| \leq 1$ for all $\boldsymbol{x}$ and $\boldsymbol{h}$. Therefore,

$$|r_a(\boldsymbol{x})| \leq \int_\Omega Cp(\boldsymbol{h}) \, \mathrm{d}\boldsymbol{h} = C, \tag{25}$$

since $p$ integrates to one as a probability density or mass function on $\Omega$.

**Concentration via Hoeffding.** Recall that the empirical mixture covariate measure is

$$\mu = \frac{1}{n}\sum_{i=1}^n \delta_{\boldsymbol{x}_i}, \quad \langle r_a, \mu \rangle_* = \frac{1}{n}\sum_{i=1}^n r_a(\boldsymbol{x}_i). \tag{26}$$

Thus, $\langle r_a, \mu \rangle_*$ is the empirical average of the outcome regression function over the full sample. By the law of total expectation,

$$\mathbb{E}[Y_a] = \mathbb{E}[\mathbb{E}[Y_a \mid \boldsymbol{X}]] = \mathbb{E}[r_a(\boldsymbol{X})], \quad a \in \{0, 1\}. \tag{27}$$

Therefore, the deviation of the empirical regression average from its population counterpart can be written as

$$\langle r_a, \mu \rangle_* - \mathbb{E}[Y_a] = \frac{1}{n}\sum_{i=1}^n r_a(\boldsymbol{X}_i) - \mathbb{E}[r_a(\boldsymbol{X})]. \tag{28}$$

Since $|r_a(\boldsymbol{X}_i)| \leq C$ almost surely, Hoeffding's inequality implies that, for any $\delta_a \in (0, 1)$,

$$\mathbb{P}\left(|\langle r_a, \mu \rangle_* - \mathbb{E}[Y_a]| \geq C\sqrt{\frac{2\log(2/\delta_a)}{n}}\right) \leq \delta_a. \tag{29}$$

Setting $\delta_0 = \delta_1 = \delta/2$ and applying a union bound over $a \in \{0, 1\}$, we obtain that, with probability at least $1 - \delta$,

$$|\langle r_a, \mu \rangle_* - \mathbb{E}[Y_a]| \leq C\sqrt{\frac{2\log(4/\delta)}{n}}, \quad a \in \{0, 1\}. \tag{30}$$

Finally, combining the two deviations by the triangle inequality gives

$$\begin{aligned}|e_{\text{sam}}| &= \left|\left(\langle r_1, \mu \rangle_* - \mathbb{E}[Y_1]\right) - \left(\langle r_0, \mu \rangle_* - \mathbb{E}[Y_0]\right)\right| \\ &\leq |\langle r_1, \mu \rangle_* - \mathbb{E}[Y_1]| + |\langle r_0, \mu \rangle_* - \mathbb{E}[Y_0]| \leq 2C\sqrt{\frac{2\log(4/\delta)}{n}},\end{aligned} \tag{31}$$

which completes the proof. $\square$

# D. Proof of Lemma 4.7

For convenience, we restate Lemma 4.7 under the RKHS-ball specialization introduced in the main paper. Compared with the main-text formulation, this restatement makes explicit the additional kernel, RKHS, and random-feature details needed for the proof. All notation introduced in this section is local to the proof.

**Lemma D.1** ($\mathcal{F}$-discrepancy upper bound for RKHS balls)**.** *Let $k$ be a positive definite kernel with $k(\boldsymbol{x}, \boldsymbol{x}) \leq 1$ and RKHS $\mathcal{H}_k$. Let $\mathcal{F} := \{f \in \mathcal{H}_k : \|f\|_{\mathcal{H}_k} \leq C\}$. Assume a random-feature representation $k(\boldsymbol{x}, \boldsymbol{x}') = \mathbb{E}_{\boldsymbol{h} \sim p}[\psi(\boldsymbol{x}; \boldsymbol{h})\psi(\boldsymbol{x}'; \boldsymbol{h})]$ with $|\psi(\boldsymbol{x}; \boldsymbol{h})| \leq 1$, and draw $\boldsymbol{h}_1, \ldots, \boldsymbol{h}_d \overset{\text{i.i.d.}}{\sim} p$ to form $\boldsymbol{\phi}(\boldsymbol{x}) = \frac{1}{\sqrt{d}}(\psi(\boldsymbol{x}; \boldsymbol{h}_1), \ldots, \psi(\boldsymbol{x}; \boldsymbol{h}_d))^\top$, $\Phi_d(\nu) = \int \boldsymbol{\phi}(\boldsymbol{x}) \, d\nu(\boldsymbol{x})$. Then for any $\delta \in (0, 1)$, with probability at least $1 - \delta$,*

$$d_{\mathcal{F}}(\nu, \nu') \leq C \left( \|\Phi_d(\nu) - \Phi_d(\nu')\|_2 + \left( \frac{8 \log(2/\delta)}{d} \right)^{1/4} \right). \tag{32}$$

*Proof.* Let $\Delta := \nu - \nu'$. By the reproducing property and RKHS duality, the discrepancy over the radius-$C$ RKHS ball can be written as

$$d_{\mathcal{F}}(\nu, \nu') = \sup_{\|f\|_{\mathcal{H}_k} \leq C} \left| \int f \, d\Delta \right| = C \|\mu_k(\nu) - \mu_k(\nu')\|_{\mathcal{H}_k}, \tag{33}$$

where $\mu_k(\nu) := \int k(\boldsymbol{x}, \cdot) \, d\nu(\boldsymbol{x})$ denotes the kernel mean embedding of $\nu$. It therefore suffices to approximate the RKHS distance between the two mean embeddings. We analyze the squared RKHS distance

$$D^2 := \|\mu_k(\nu) - \mu_k(\nu')\|_{\mathcal{H}_k}^2 = \iint k(\boldsymbol{x}, \boldsymbol{y}) \, d\Delta(\boldsymbol{x}) \, d\Delta(\boldsymbol{y}). \tag{34}$$

Using the random-feature representation of the kernel and Fubini's theorem, we obtain

$$D^2 = \iint \mathbb{E}_{\boldsymbol{h}} \big[ \psi(\boldsymbol{x}; \boldsymbol{h}) \psi(\boldsymbol{y}; \boldsymbol{h}) \big] \, d\Delta(\boldsymbol{x}) \, d\Delta(\boldsymbol{y}) = \mathbb{E}_{\boldsymbol{h}} \left[ \left( \int \psi(\boldsymbol{x}; \boldsymbol{h}) \, d\Delta(\boldsymbol{x}) \right)^2 \right]. \tag{35}$$

For a fixed $\boldsymbol{h}$, define the random variable $Z(\boldsymbol{h}) := \left( \int \psi(\boldsymbol{x}; \boldsymbol{h}) \, d\Delta(\boldsymbol{x}) \right)^2$. Since $|\psi| \leq 1$ and $\Delta$ is the difference of two probability measures, we have $\left| \int \psi \, d\Delta \right| \leq \int |\psi| \, d(\nu + \nu') \leq 2$. Thus, $Z(\boldsymbol{h})$ is bounded as $0 \leq Z(\boldsymbol{h}) \leq 4$.

The empirical squared distance in the random-feature space is precisely the sample average of these variables:

$$\|\Phi_d(\nu) - \Phi_d(\nu')\|_2^2 = \sum_{i=1}^{d} \left( \frac{1}{\sqrt{d}} \int \psi(\boldsymbol{x}; \boldsymbol{h}_i) \, d\Delta(\boldsymbol{x}) \right)^2 = \frac{1}{d} \sum_{i=1}^{d} Z(\boldsymbol{h}_i). \tag{36}$$

Since $\boldsymbol{h}_i \overset{\text{i.i.d.}}{\sim} p$, Hoeffding's inequality for bounded variables in $[0, 4]$ implies that, with probability at least $1 - \delta$,

$$\left| \|\Phi_d(\nu) - \Phi_d(\nu')\|_2^2 - \mathbb{E}[Z(\boldsymbol{h})] \right| \leq (4 - 0) \sqrt{\frac{\log(2/\delta)}{2d}} = \sqrt{\frac{8 \log(2/\delta)}{d}}. \tag{37}$$

Substituting $\mathbb{E}[Z(\boldsymbol{h})] = D^2 = (d_{\mathcal{F}}(\nu, \nu')/C)^2$ into the preceding inequality gives

$$\left( \frac{d_{\mathcal{F}}(\nu, \nu')}{C} \right)^2 \leq \|\Phi_d(\nu) - \Phi_d(\nu')\|_2^2 + \sqrt{\frac{8 \log(2/\delta)}{d}}. \tag{38}$$

Taking square roots and using $\sqrt{a + b} \leq \sqrt{a} + \sqrt{b}$ for $a, b \geq 0$ yields

$$\frac{d_{\mathcal{F}}(\nu, \nu')}{C} \leq \|\Phi_d(\nu) - \Phi_d(\nu')\|_2 + \left( \frac{8 \log(2/\delta)}{d} \right)^{1/4}. \tag{39}$$

Multiplying both sides by $C$ gives the stated bound and completes the proof. $\qquad\square$

## E. Proof of Corollary 4.8

For convenience, we restate Corollary 4.8. All notation introduced in this part is local and used only for the proof.

**Corollary E.1** (Coreset-induced error bound). *For any $\delta \in (0, 1)$, with probability at least $1 - \delta$,*

$$|e_{\text{cor}}| \leq 2\, d_{\mathcal{F}}(\mu_u, \mu) \leq 2C\varepsilon + 2C \left( \frac{8 \log(2/\delta)}{d} \right)^{1/4}. \tag{40}$$

*Proof.* Recall that $e_{\text{cor}} := \langle r_1 - r_0, \mu_u - \mu \rangle_*$. By Assumption 4.1, $r_0, r_1 \in \mathcal{F}$. Since $\mathcal{F}$ is symmetric and scales linearly with its coefficient bound, the difference $r_1 - r_0$ belongs to the enlarged class $\mathcal{F}_{2C}$, defined as the same function class with coefficient bound $2C$. This allows us to control $r_1 - r_0$ using the same discrepancy, up to a factor of two. Consequently,

$$\sup_{g \in \mathcal{F}_{2C}} \left| \int g\, \mathrm{d}(\mu_u - \mu) \right| = 2 \sup_{f \in \mathcal{F}} \left| \int f\, \mathrm{d}(\mu_u - \mu) \right| = 2\, d_{\mathcal{F}}(\mu_u, \mu), \tag{41}$$

where the first equality uses the linear scaling of the coefficient bound in the definition of $\mathcal{F}$.

Therefore,

$$|e_{\text{cor}}| = \left| \int_{\mathcal{X}} (r_1 - r_0)(\boldsymbol{x})\, \mathrm{d}(\mu_u - \mu)(\boldsymbol{x}) \right| \leq 2\, d_{\mathcal{F}}(\mu_u, \mu), \tag{42}$$

which proves the first inequality in Eq. (40).

It remains to upper bound $d_{\mathcal{F}}(\mu_u, \mu)$. Applying Lemma 4.7 with $\nu = \mu_u$ and $\nu' = \mu$ gives, with probability at least $1 - \delta$ over the random feature draw,

$$d_{\mathcal{F}}(\mu_u, \mu) \leq C \left( \left\| \Phi_d(\mu_u) - \Phi_d(\mu) \right\|_2 + \left( \frac{8 \log(2/\delta)}{d} \right)^{1/4} \right). \tag{43}$$

By Proposition 4.4 and Theorem 4.3, the maintained coreset mixture satisfies the feature-embedding approximation bound

$$\left\| \Phi_d(\mu_u) - \Phi_d(\mu) \right\|_2 \leq \varepsilon. \tag{44}$$

Substituting this into Eq. (43) yields

$$d_{\mathcal{F}}(\mu_u, \mu) \leq C \left( \varepsilon + \left( \frac{8 \log(2/\delta)}{d} \right)^{1/4} \right). \tag{45}$$

Combining this bound with Eq. (42) gives

$$|e_{\text{cor}}| \leq 2C\varepsilon + 2C \left( \frac{8 \log(2/\delta)}{d} \right)^{1/4}. \tag{46}$$

This completes the proof. $\square$

## F. Proof of Lemma 4.10

For convenience, we restate Lemma 4.10. All notation introduced in this part is local and used only for the proof.

**Lemma F.1** (Outcome noise error). *Under Assumption 4.9, for any $\delta \in (0, 1)$, with probability at least $1 - \delta$,*

$$|e_{\text{noi}}| \leq \sigma \sqrt{2 \log(4/\delta)} \left( \sqrt{\sum_{j \in \mathcal{S}_t} z_{t,j}^2} + \sqrt{\sum_{j \in \mathcal{S}_c} z_{c,j}^2} \right).$$

*Proof.* Recall that $e_{\text{noi}} = \langle y_1 - r_1, \mu_{t,uw} \rangle_* - \langle y_0 - r_0, \mu_{c,uw} \rangle_*$. Since $\mu_{t,uw}$ and $\mu_{c,uw}$ are supported on the treated and control coresets with final normalized weights $\boldsymbol{z}_t$ and $\boldsymbol{z}_c$, respectively, we can write

$$e_{\text{noi}} = \sum_{j \in \mathcal{S}_t} z_{t,j}\, \eta_{1,j} - \sum_{j \in \mathcal{S}_c} z_{c,j}\, \eta_{0,j}, \tag{47}$$

where $\eta_{a,j} := y_{a,j} - r_a(\boldsymbol{x}_j)$ denotes the sampled outcome noise. Importantly, the coreset support and weights depend on the covariates, random features, and balancing procedure, but not on the outcome noises. Thus, after conditioning on the covariates and the resulting weights, the weights can be treated as fixed.

**Step 1: Weighted sub-Gaussian sums.** Fix $a \in \{0, 1\}$ and consider a weighted sum over an index set $\mathcal{S}$:

$$T := \sum_{j \in \mathcal{S}} z_j \, \eta_{a,j}. \tag{48}$$

Conditioning on the covariates and the resulting weights $\{z_j\}_{j \in \mathcal{S}}$, the weights are fixed and the noises $\{\eta_{a,j}\}_{j \in \mathcal{S}}$ are independent, mean-zero, and $\sigma^2$-sub-Gaussian by Assumption 4.9. Therefore, for any $\kappa \in \mathbb{R}$,

$$\mathbb{E}[\exp(\kappa T) \mid \{\boldsymbol{x}_j, z_j\}_{j \in \mathcal{S}}] = \prod_{j \in \mathcal{S}} \mathbb{E}[\exp(\kappa z_j \eta_{a,j}) \mid \{\boldsymbol{x}_j, z_j\}_{j \in \mathcal{S}}] \le \prod_{j \in \mathcal{S}} \exp\left(\frac{\kappa^2 z_j^2 \sigma^2}{2}\right) = \exp\left(\frac{\kappa^2 \sigma^2}{2} \sum_{j \in \mathcal{S}} z_j^2\right). \tag{49}$$

Thus, conditional on the covariates and weights, $T$ is sub-Gaussian with variance proxy $\sigma^2 \sum_{j \in \mathcal{S}} z_j^2$. The standard sub-Gaussian tail bound gives, for any $t > 0$,

$$\mathbb{P}(|T| \ge t \mid \{\boldsymbol{x}_j, z_j\}_{j \in \mathcal{S}}) \le 2 \exp\left(-\frac{t^2}{2\sigma^2 \sum_{j \in \mathcal{S}} z_j^2}\right). \tag{50}$$

Since this conditional tail bound holds for every realization of the covariates and weights, we can average both sides over their randomness. By the tower property, the same probability bound holds unconditionally.

**Step 2: Apply to treated and control sums and union bound.** Apply Eq. (50) to the treated sum $T_t := \sum_{j \in \mathcal{S}_t} z_{t,j} \eta_{1,j}$ and the control sum $T_c := \sum_{j \in \mathcal{S}_c} z_{c,j} \eta_{0,j}$. Setting $t_t := \sigma \sqrt{2 \log(4/\delta)} \sqrt{\sum_{j \in \mathcal{S}_t} z_{t,j}^2}$ and using Eq. (50) gives

$$\mathbb{P}\left(|T_t| \ge \sigma \sqrt{2 \log(4/\delta)} \sqrt{\sum_{j \in \mathcal{S}_t} z_{t,j}^2}\right) \le \frac{\delta}{2}. \tag{51}$$

Similarly,

$$\mathbb{P}\left(|T_c| \ge \sigma \sqrt{2 \log(4/\delta)} \sqrt{\sum_{j \in \mathcal{S}_c} z_{c,j}^2}\right) \le \frac{\delta}{2}. \tag{52}$$

By a union bound, with probability at least $1 - \delta$, both inequalities fail to occur simultaneously, i.e., $|T_t| \le t_t$ and $|T_c| \le t_c$.

**Step 3: Conclude the bound for $e_{\mathrm{noi}}$.** On this event, using $e_{\mathrm{noi}} = T_t - T_c$ and the triangle inequality,

$$|e_{\mathrm{noi}}| \le |T_t| + |T_c| \le \sigma \sqrt{2 \log(4/\delta)} \left(\sqrt{\sum_{j \in \mathcal{S}_t} z_{t,j}^2} + \sqrt{\sum_{j \in \mathcal{S}_c} z_{c,j}^2}\right). \tag{53}$$

This completes the proof. $\qquad\square$

## G. Proof of Lemma 4.11

For convenience, we restate Lemma 4.11. All notation introduced in this part is local and used only for the proof.

**Lemma G.1** (Balancing error bound). *For any $\delta \in (0, 1)$, with probability at least $1 - \delta$,*

$$|e_{\mathrm{bal}}| \le C \left(2 \left(\frac{8 \log(4/\delta)}{d}\right)^{1/4} + \|\Phi_d(\mu_{t,uw}) - \Phi_d(\mu_u)\|_2 + \|\Phi_d(\mu_{c,uw}) - \Phi_d(\mu_u)\|_2\right). \tag{54}$$

*Proof.* Recall that $e_{\mathrm{bal}} := \langle r_1, \mu_{t,uw} - \mu_u \rangle_* - \langle r_0, \mu_{c,uw} - \mu_u \rangle_*$. This term measures the remaining imbalance after reweighting the treated and control coresets toward the coreset-induced mixture measure $\mu_u$. By the triangle inequality,

$$|e_{\mathrm{bal}}| \le |\langle r_1, \mu_{t,uw} - \mu_u \rangle_*| + |\langle r_0, \mu_{c,uw} - \mu_u \rangle_*|. \tag{55}$$

Under Assumption 4.1, we have $r_0, r_1 \in \mathcal{F}$. By the definition of the $\mathcal{F}$-discrepancy in Definition 4.6, for any $f \in \mathcal{F}$ and any two measures $\nu, \nu'$,

$$|\langle f, \nu - \nu' \rangle_*| \leq \sup_{g \in \mathcal{F}} |\langle g, \nu - \nu' \rangle_*| = d_{\mathcal{F}}(\nu, \nu'). \tag{56}$$

Applying this inequality with $(f, \nu, \nu') = (r_1, \mu_{t,uw}, \mu_u)$ and $(f, \nu, \nu') = (r_0, \mu_{c,uw}, \mu_u)$ and substituting into Eq. (55) yields

$$|e_{\text{bal}}| \leq d_{\mathcal{F}}(\mu_{t,uw}, \mu_u) + d_{\mathcal{F}}(\mu_{c,uw}, \mu_u). \tag{57}$$

It remains to upper bound the two discrepancy terms using finite-dimensional random feature embeddings. Apply Lemma 4.7 to $(\nu, \nu') = (\mu_{t,uw}, \mu_u)$ with confidence level $\delta/2$. Then, with probability at least $1 - \delta/2$,

$$d_{\mathcal{F}}(\mu_{t,uw}, \mu_u) \leq C \left( \left\| \Phi_d(\mu_{t,uw}) - \Phi_d(\mu_u) \right\|_2 + \left( \frac{8 \log(4/\delta)}{d} \right)^{1/4} \right). \tag{58}$$

Similarly, applying Lemma 4.7 to $(\nu, \nu') = (\mu_{c,uw}, \mu_u)$ with confidence level $\delta/2$ yields that, with probability at least $1 - \delta/2$,

$$d_{\mathcal{F}}(\mu_{c,uw}, \mu_u) \leq C \left( \left\| \Phi_d(\mu_{c,uw}) - \Phi_d(\mu_u) \right\|_2 + \left( \frac{8 \log(4/\delta)}{d} \right)^{1/4} \right). \tag{59}$$

By a union bound, Eq. (58) and Eq. (59) hold simultaneously with probability at least $1 - \delta$. Combining these two bounds with Eq. (57) gives

$$|e_{\text{bal}}| \leq C \left( \left\| \Phi_d(\mu_{t,uw}) - \Phi_d(\mu_u) \right\|_2 + \left\| \Phi_d(\mu_{c,uw}) - \Phi_d(\mu_u) \right\|_2 + 2 \left( \frac{8 \log(4/\delta)}{d} \right)^{1/4} \right). \tag{60}$$

This proves the desired balancing error bound and completes the proof. $\qquad\square$

## H. Derivation of Eq. (17)

We provide the derivation of Eq. (17) by combining the four bounds for the error decomposition. Recall that

$$\widehat{\text{ATE}}_{u,w} - \text{ATE} = e_{\text{sam}} + e_{\text{cor}} + e_{\text{noi}} + e_{\text{bal}}. \tag{61}$$

By the triangle inequality, $|\widehat{\text{ATE}}_{u,w} - \text{ATE}| \leq |e_{\text{sam}}| + |e_{\text{cor}}| + |e_{\text{noi}}| + |e_{\text{bal}}|$. We then apply Lemma 4.5, Corollary 4.8, Lemma 4.10, and Lemma 4.11 with confidence level $\delta/4$ each, and take a union bound. Thus, with probability at least $1 - \delta$,

$$\left| \widehat{\text{ATE}}_{u,w} - \text{ATE} \right| \leq 2C \sqrt{\frac{2 \log(16/\delta)}{n}} + 2C\varepsilon + 2C \left( \frac{8 \log(8/\delta)}{d} \right)^{1/4}$$

$$+ \sigma \sqrt{2 \log(16/\delta)} \left( \sqrt{\sum_{j \in \mathcal{S}_t} z_{t,j}^2} + \sqrt{\sum_{j \in \mathcal{S}_c} z_{c,j}^2} \right)$$

$$+ C \left( 2 \left( \frac{8 \log(16/\delta)}{d} \right)^{1/4} + \left\| \Phi_d(\mu_{t,uw}) - \Phi_d(\mu_u) \right\|_2 + \left\| \Phi_d(\mu_{c,uw}) - \Phi_d(\mu_u) \right\|_2 \right). \tag{62}$$

The final line contains the only directly algorithm-controlled component, namely the feature-mean matching discrepancy induced by the balancing weights. By the inequality $a + b \leq \sqrt{2(a^2 + b^2)}$ for $a, b \geq 0$, we have

$$\left\| \Phi_d(\mu_{t,uw}) - \Phi_d(\mu_u) \right\|_2 + \left\| \Phi_d(\mu_{c,uw}) - \Phi_d(\mu_u) \right\|_2 \leq \sqrt{2 \mathcal{L}_\Phi},$$

where $\mathcal{L}_\Phi$ is exactly the covariate-balancing objective minimized in Eq. (14). This yields the summarized rate in Eq. (17); equivalently, up to logarithmic factors in $\delta$ and problem-dependent constants,

$$\left| \widehat{\text{ATE}}_{u,w} - \text{ATE} \right| = \mathcal{O} \left( \varepsilon + \frac{1}{d^{1/4}} + \frac{1}{\sqrt{n}} \right) + \mathcal{O} \left( \sqrt{\mathcal{L}_\Phi} \right) + \mathcal{O} \left( \sqrt{\sum_{j \in \mathcal{S}_t} z_{t,j}^2} + \sqrt{\sum_{j \in \mathcal{S}_c} z_{c,j}^2} \right). \tag{63}$$

# I. Details of Experimental Setup and Analysis of Results

**Implementation details.**   For the baseline methods, we use the publicly available R implementation of Luo et al. (2024)[2] with default settings. Following their recommendations, we use the `Gaussian` family for continuous outcomes and the `binary` family for binary outcomes. For FCore, we set the feature-map parameter $\gamma$ in $\phi(\cdot)$ to 3.0, use $d = 200$ random features across all experiments, and set the default optimization learning rate to 0.03.

## I.1. Coreset Visualization via Kernel Density Estimation

**Data generation.**   To qualitatively assess the fidelity of the proposed streaming coreset, we construct a two-dimensional synthetic dataset from an imbalanced Gaussian mixture model. Specifically, we generate $n = 10,000$ samples from a three-component mixture with parameters

$$\begin{aligned}
\boldsymbol{\pi} &= (0.85,\, 0.10,\, 0.05), \\
\boldsymbol{\mu}_1 &= (0,0), \quad \boldsymbol{\mu}_2 = (4,4), \quad \boldsymbol{\mu}_3 = (-4,3), \\
\boldsymbol{\Sigma}_1 &= \begin{bmatrix} 1.0 & 0.6 \\ 0.6 & 1.2 \end{bmatrix}, \quad \boldsymbol{\Sigma}_2 = \begin{bmatrix} 0.4 & 0 \\ 0 & 0.4 \end{bmatrix}, \quad \boldsymbol{\Sigma}_3 = \begin{bmatrix} 0.2 & 0 \\ 0 & 0.8 \end{bmatrix}.
\end{aligned} \tag{64}$$

All results are generated with a fixed random seed for reproducibility.

**Streaming coreset construction.**   We construct the streaming coreset using random Fourier features with $d = 200$ features and bucket size $m = 400$. The coreset is maintained in a single pass over the data stream according to Algorithm 1, so that only a compressed weighted summary is stored throughout the process rather than the full dataset. This setup reflects the intended streaming regime, where the memory budget is fixed and the data are processed sequentially. To assess the benefit of the proposed discrepancy-aware signing strategy, we compare against a random sketch baseline that uses random signs $\sigma_i \sim \text{Unif}\{\pm 1\}$ during compression.

**Visualization results.**   After processing the entire stream, we compare the kernel density estimate (KDE) induced by the coreset with that computed from the full dataset. The KDEs are evaluated on a $200 \times 200$ grid with padding 2.0 around the data range. This comparison provides a visual diagnostic of whether the compressed summary preserves the global density shape as well as the smaller mixture components. As shown in Fig. 2, the proposed coreset closely preserves the multimodal structure of the underlying distribution under a fixed memory budget. In particular, it captures both the dominant component and the smaller low-density components, indicating that the compression does not simply focus on high-mass regions.

## I.2. Additional Details on Synthetic Experiments

In this subsection, we provide additional details on the experiments conducted on two synthetic datasets. The synthetic streams are designed to contain batch-level distributional changes, so that the covariate composition and/or outcome-generating mechanism varies across batches. This deliberately creates non-uniform temporal structure within the stream and allows us to evaluate whether each method can remain accurate under distribution shifts.

**Evaluation protocol.**   We evaluate streaming performance using a prefix protocol. After observing the first $t$ batches, we aggregate all samples from batches $\{1, \ldots, t\}$ and estimate the ATE using only information available up to time $t$. This mimics an anytime streaming setting, where the estimator must be updated as new batches arrive rather than retrained on a fixed static dataset. We report mean absolute error (MAE) and related metrics as functions of $t$, which reflect each method's ability to track the target ATE under the imposed batch-level shifts.

### I.2.1. HETEROGENEOUS OUTCOME SHIFT

**Data generation process.**   We generate covariates $X = (X_1, \ldots, X_6) \in \mathbb{R}^6$ with independent components $X_j \sim \text{Unif}(-2, 2)$. Let $m_1 = \max\{X_1, X_3, X_6\}$ and $m_2 = \max\{X_2, X_4, X_5\}$. Treatment assignment follows a nonlinear propensity model. Specifically, we first compute

$$\mu_1(X) = 0.9\tanh(0.8m_1) + 0.6\sin(X_2 + 0.5X_4) + 0.4\cos(1.5X_5) + 0.25X_3 - 0.2X_4^2 + 0.35\log(1 + m_2^2) - 2.2, \tag{65}$$

---

[2]`https://github.com/luolsph/OnlineCausal`

then sample $a' \sim \mathcal{N}(\mu_1(X), 0.5)$ and set $A \sim \text{Ber}(\sigma(a'))$, where $\sigma(\cdot)$ denotes the logistic function.

To model shifts in the outcome-generating mechanism, we generate a stream of $B = 5$ batches, each containing 2000 samples. Within batch $b \in \{1, \ldots, 5\}$, outcomes are drawn as

$$Y \sim \mathcal{N}\big(g_b(X) + A\,\tau_b(X),\, 0.1\big), \tag{66}$$

where the baseline outcome function $g_b(X)$ increases in complexity across batches:

$$\begin{aligned}
g_1(X) &= 2.6(2X_4 - 1)^2, \\
g_2(X) &= 2.6(2X_4 - 1)^2 + 1.8(X_1 - 2)^2, \\
g_3(X) &= 2.6(2X_4 - 1)^2 + 1.8(X_1 - 2)^2 + 0.8X_3^2, \\
g_4(X) &= 2.6(2X_4 - 1)^2 + 1.8(X_1 - 2)^2 + 0.8X_3^2 + \frac{0.7\sin(X_2)}{1 + X_6^2}, \\
g_5(X) &= g_4(X),
\end{aligned} \tag{67}$$

and the heterogeneous treatment effect rotates across covariates to induce nonstationarity:

$$\tau_b(X) = 3\cos(2X_b) + 2\sin(X_b), \quad b \in \{1, \ldots, 5\}. \tag{68}$$

For each batch $b$, the ground-truth average treatment effect satisfies $\mathbb{E}[\tau_b(X)] = \frac{3}{4}\sin(4)$ under $X_b \sim \text{Unif}(-2, 2)$. Finally, we standardize covariates within each batch using the pooled treated and control samples before applying all methods.

**Results and analysis.** Figure 6 summarizes MAE across stream prefixes under heterogeneous outcome shift. The model-based baselines, including G-comp, IPTW, and AIPTW, exhibit consistently large errors across all prefixes, reflecting their sensitivity to non-stationary outcome-generating processes and the accumulation of model misspecification over time. In contrast, FCore achieves substantially lower MAE throughout the stream and remains stable as the prefix length increases, indicating strong robustness to heterogeneous and evolving outcome mechanisms.

We further observe that FCore is resilient to aggressive coreset compression. As the retention ratio $\rho$ decreases, the estimation error increases gradually rather than abruptly, suggesting that the feature coreset preserves the statistics relevant for covariate balancing even under severe memory constraints. Increasing $\rho$ leads to monotonic performance improvements and approaches the full-retention case ($\rho = 1.0$), consistent with the expected accuracy–memory trade-off. Overall, these results demonstrate that FCore enables reliable ATE estimation under outcome shift while operating with limited memory, outperforming model-based online baselines across all prefixes.

### I.2.2. KANG–SCHAFER TREATMENT SHIFT

**Data generation process.** We consider a Kang–Schafer-style synthetic benchmark with latent confounding and batch-dependent treatment assignment shifts. For each batch $b$, we generate latent covariates $Z = (Z_1, \ldots, Z_4) \in \mathbb{R}^4$ with independent components $Z_j \sim \mathcal{N}(0, 1)$. The observed covariates $X \in \mathbb{R}^4$ are obtained through the standard nonlinear Kang–Schafer transformation:

$$X_1 = \exp(Z_1/2), \quad X_2 = \frac{Z_2}{1 + \exp(Z_1)} + 10, \quad X_3 = \big(Z_1 Z_3/25 + 0.6\big)^3, \quad X_4 = (Z_2 + Z_4 + 20)^2. \tag{69}$$

Treatment assignment follows a nonlinear propensity model defined on the latent variables:

$$\pi_b(Z) = \sigma\big(Z^\top \beta + \nu_b\big), \quad \beta = (-1.0,\, 0.5,\, -0.25,\, -0.1)^\top, \tag{70}$$

where $\sigma(\cdot)$ denotes the logistic function. To induce temporal shifts in the treatment assignment mechanism, we vary the intercept $\nu_b$ across batches as

$$\nu_b \in \{0.0,\, -0.5,\, -1.0,\, -1.5,\, -2.0\}, \tag{71}$$

which progressively decreases the treatment probability over time. Treatment is then sampled as $A \sim \text{Ber}(\pi_b(Z))$.

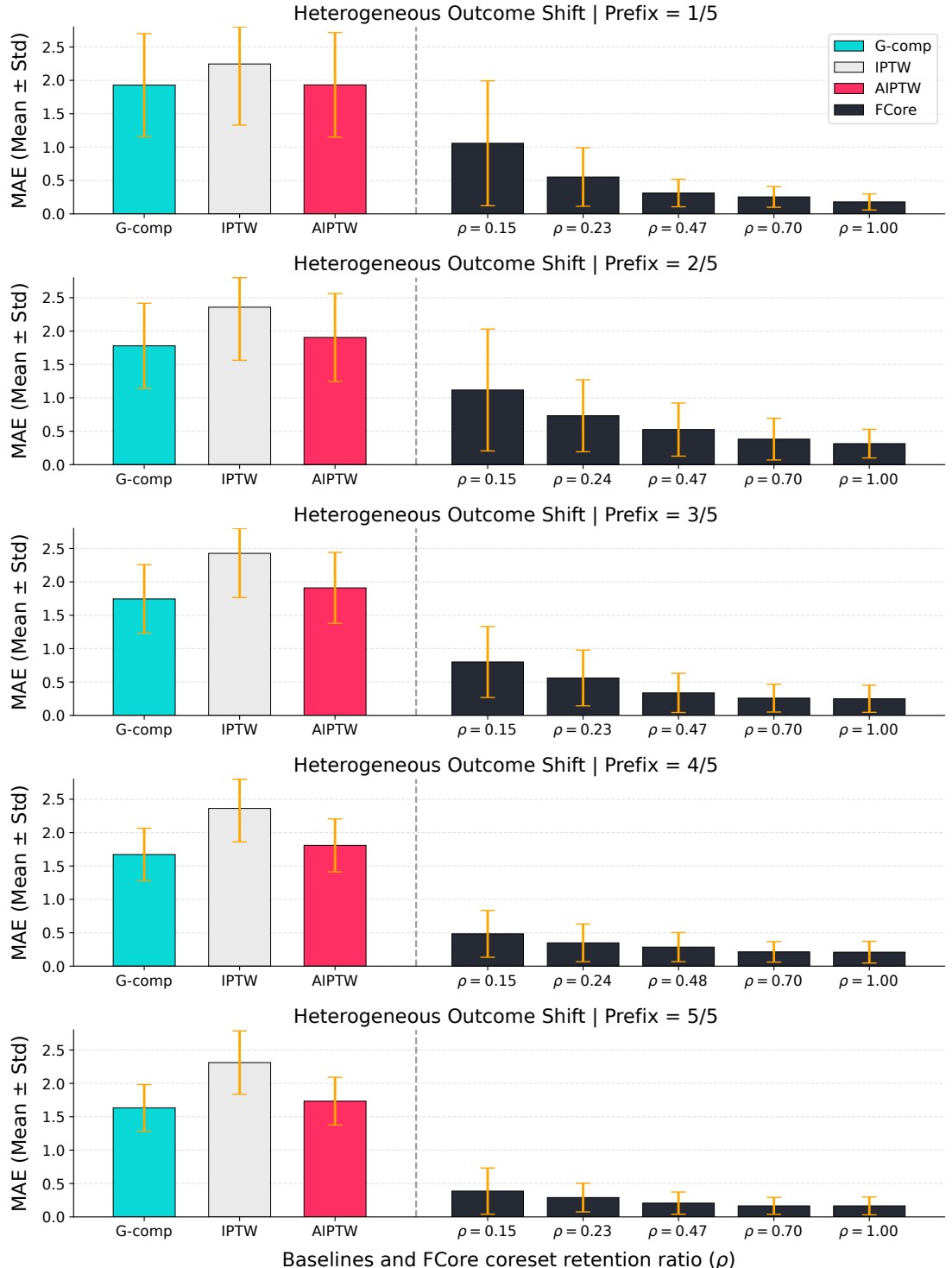

*Figure 6.* Performance across different prefixes under Heterogeneous Outcome Shift.

Outcomes are generated without a treatment effect so that the ground-truth average treatment effect (ATE) is zero for all batches. Specifically, outcomes are sampled as

$$Y = \mu + Z^\top \gamma + \varepsilon, \quad \mu = 210, \quad \gamma = (27.4,\ 13.7,\ 13.7,\ 13.7)^\top, \quad \varepsilon \sim \mathcal{N}(0,1). \tag{72}$$

implying ATE $= 0$ regardless of the batch index $b$. Thus, this design isolates treatment assignment shift while keeping the outcome-generating process stationary.

Each batch contains $n = 2000$ samples, and we generate a stream of $B = 5$ batches. Before applying all methods, observed covariates are standardized within each batch using the pooled treated and control samples.

**Results and analysis.** Figure 7 reports the MAE across stream prefixes under the Kang–Schafer treatment shift setting, where the treatment assignment mechanism changes over time while the outcome-generating process remains stationary and the ground-truth ATE is zero. We observe that propensity-score-based baselines, especially IPTW and AIPTW, exhibit large estimation errors and high variability across prefixes. This is consistent with their sensitivity to changes in treatment prevalence: as the propensity intercept shifts across batches, previously learned propensity models become less reliable and the resulting inverse-probability weights can become unstable. G-comp is comparatively more stable, but still incurs non-negligible bias due to latent confounding and model misspecification.

In contrast, FCore consistently achieves near-zero MAE across prefixes, demonstrating strong robustness to evolving treatment assignment mechanisms. It also remains stable under aggressive coreset compression, with only mild degradation as the retention ratio $\rho$ decreases. As $\rho$ increases, the estimation error further decreases and approaches the full-retention case ($\rho = 1.0$), reflecting the expected accuracy–memory trade-off. Overall, these results highlight the advantage of directly learning balancing weights without relying on parametric propensity score models, enabling reliable ATE estimation under treatment shift while operating with bounded memory.

### I.3. Additional Details on Real-World Experiments

To evaluate the effectiveness of the proposed method in real-world scenarios, we consider two datasets: the Infant Health and Development Program (IHDP) dataset and a COVID-19 vaccine safety monitoring dataset. We describe the experimental settings for both datasets below.

**IHDP**: The Infant Health and Development Program (IHDP) dataset studies the causal effect of specialist home visits on infants' subsequent cognitive test scores. Each sample contains 25 covariates describing characteristics of both the child and the mother. Following prior work, we adopt Setting A from the NPCI package (Dorie, 2016). All covariates are continuous and are standardized prior to analysis. The dataset consists of 10 data files, each containing 747 samples, for a total of 7,470 observations. We treat each file as one batch in the streaming setting. These 10 files correspond to independent replications under the same simulation design: they share the same covariates from the original IHDP study, but differ due to randomness in treatment assignment (via a propensity-score-based non-random mechanism) and outcome noise. As a result, treatment indicators and outcomes vary across batches, while covariates remain fixed.

**COVID-19**: We demonstrate the efficacy of our proposed methods through a proof-of-concept study focused on near real-time COVID-19 vaccine safety monitoring, utilizing data from the Vaccine Adverse Event Reporting System (VAERS). VAERS is a national early warning system co-managed by the CDC and FDA, designed to detect potential safety issues in U.S.-licensed vaccines. Sparking the largest vaccination campaign in history, the COVID-19 pandemic saw the administration of more than 400 million doses in the U.S. from December 14, 2020, to October 5, 2021. Under established protocols, healthcare providers must report all administration errors and serious adverse events to VAERS, regardless of causality. De-identified data become publicly accessible with a 4–6 week delay and are updated regularly. Consequently, VAERS serves as a vital tool for the near real-time detection of post-vaccination safety concerns. Our analysis focused on headache and fatigue, the two most prevalent post-vaccination symptoms reported to VAERS by adult COVID-19 vaccine recipients throughout the entire year of 2021. We estimated the comparative vaccine effect, defined as the risk difference between the potential outcomes of receiving the Pfizer vaccine versus the Moderna vaccine. We applied our proposed online methods to update these estimates monthly throughout the entire year (i.e., 12 data batches), adjusting for gender, age, dose number, and vaccination time (converted into the number of days since January 1, 2021). For each outcome, we determine whether a patient reported the symptom by searching the symptom-description text for relevant keywords; if such keywords are present, the patient is coded as having the corresponding symptom.

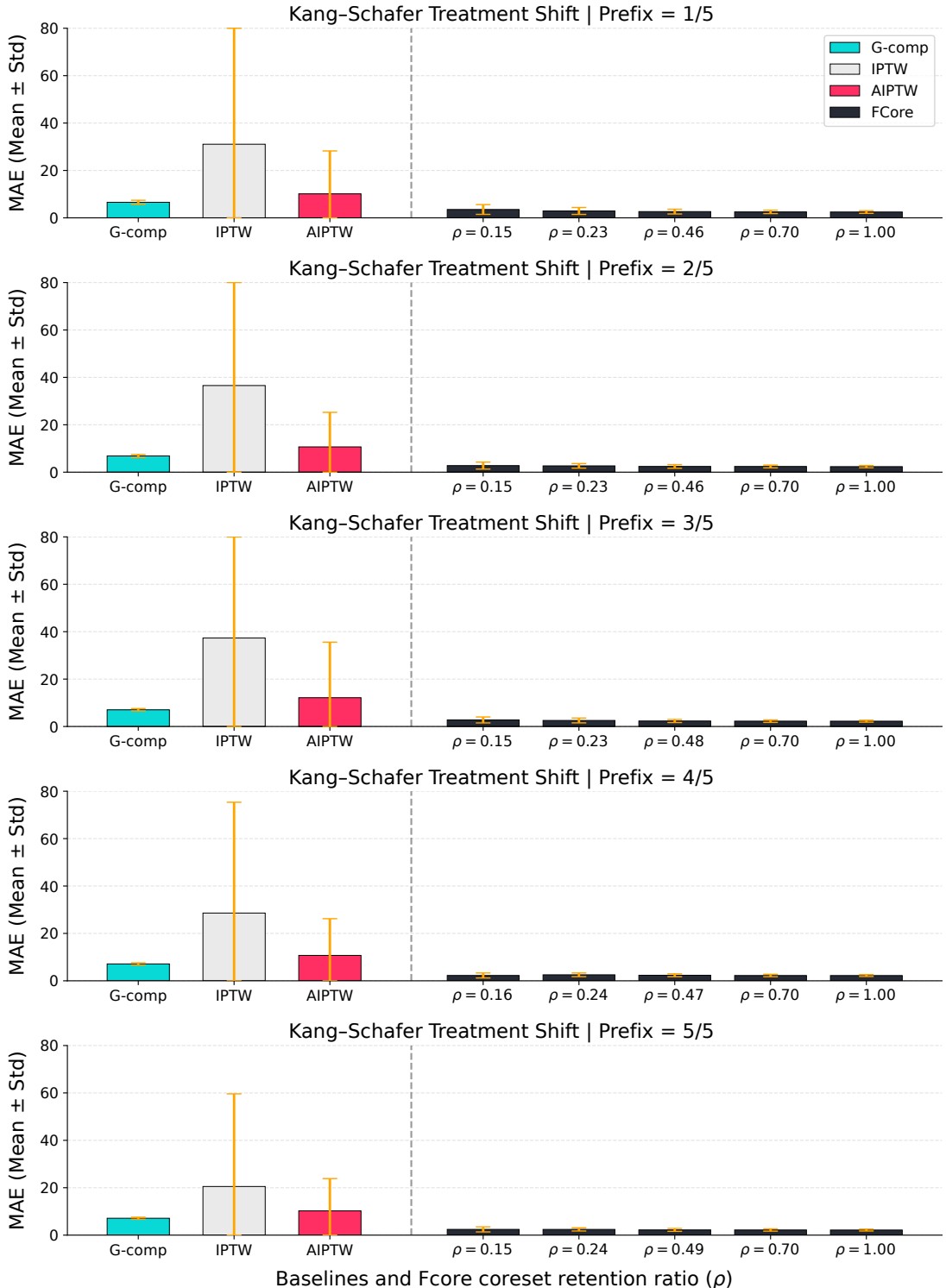

*Figure 7.* Performance across different prefixes under Kang–Schafer Treatment Shift.

# J. Additional Experiment Results

This section provides additional empirical results omitted from the main paper due to space limitations. We report runtime comparisons, sensitivity analysis for the feature-map parameter, comparisons with additional baselines, and an empirical convergence comparison for the practical optimizer.

## J.1. Runtime Comparision

We evaluate computational efficiency on the real-world COVID-19 dataset. Table 1 reports the running time of offline baselines, online baselines, and the proposed FCore method. The dataset contains approximately 600,000 samples, making single-batch processing relatively costly. Offline methods process the entire dataset at once, whereas online methods update their estimates incrementally as new data arrive. Since FCore is designed for streaming estimation, we report its runtime in the online setting.

*Table 1.* Results of running time (s) on real-world dataset.

| G-Comp (Offline) | IPTW (Offline) | AIPTW (Offline) | G-Comp (Online) | IPTW (Online) | AIPTW (Online) | Fcore (Ours) |
|---|---|---|---|---|---|---|
| 8.12 | 4.35 | 12.17 | 3.81 | 1.66 | 6.88 | 6.08 |

As expected, online implementations are generally more efficient than their offline counterparts due to their incremental updates. Among the online baselines, IPTW is the fastest, while AIPTW incurs higher computational cost. Despite achieving stronger estimation performance, FCore maintains a moderate runtime that is comparable to other online methods. These results demonstrate that FCore not only improves estimation accuracy, but also remains computationally efficient, offering a favorable balance between effectiveness and efficiency in streaming settings.

## J.2. Sensitivity Analysis for Feature-Map Parameter

We conduct a sensitivity analysis to examine the robustness of the proposed ATE estimator with respect to the feature-map parameter $\gamma$. In the main experiments, we set $\gamma = 3.0$ as the default value. To assess whether empirical performance is sensitive to this choice, we vary $\gamma$ over a wide range and report the resulting mean absolute error (MAE) under the heterogeneous outcome shift setting.

*Table 2.* Sensitivity analysis of MAE under heterogeneous outcome shift for different values of the feature-map parameter $\gamma$.

| $\gamma$ | 1.0 | 2.0 | 3.0 | 4.0 | 5.0 | 6.0 | 7.0 | 8.0 | 9.0 | 10.0 |
|---|---|---|---|---|---|---|---|---|---|---|
| MAE | 0.74 | 0.23 | 0.18 | 0.15 | 0.13 | 0.11 | 0.10 | 0.10 | 0.11 | 0.12 |

As shown in Tab. 2, the estimation performance remains stable across a broad range of $\gamma$ values. In particular, when $\gamma \in [3.0, 9.0]$, the MAE remains consistently low, ranging from 0.10 to 0.18. This indicates that the proposed estimator is not overly sensitive to the precise choice of $\gamma$ within this interval. Although performance deteriorates when $\gamma$ is too small, such as $\gamma = 1.0$, the results suggest a broad and reliable operating region for the feature-map parameter. Overall, this sensitivity analysis supports the empirical robustness of the default choice $\gamma = 3.0$.

## J.3. Comparison with Additional Baselines

**Balancing-based baselines.** We further evaluate the effectiveness of the proposed discrepancy-based coreset construction by comparing FCore with two standard balancing-based baselines, CBPS (Imai & Ratkovic, 2014) and Entropy Balancing (EB) (Hainmueller, 2012). This experiment is designed to isolate the contribution of our coreset selection strategy under the heterogeneous outcome shift setting. To ensure a fair comparison, we report results for both the full offline sample and a 20% random subsample, where the latter is chosen to match the size of the proposed coreset. The MAE at the final stream prefix is reported in Tab. 3.

As shown in Tab. 3, FCore achieves substantially lower MAE than both CBPS and EB in both sample settings. In the full offline setting, FCore reduces the MAE from 3.017 and 1.519 to 0.165 compared with CBPS and EB, respectively. Under the size-matched 20% random subsample setting, FCore also maintains a clear advantage, achieving an MAE of 0.290, while CBPS and EB obtain MAEs of 2.912 and 1.616, respectively. These results indicate that the performance improvement is not merely due to using fewer samples, but rather stems from the proposed discrepancy-based coreset construction.

*Table 3.* Comparison of MAE under heterogeneous outcome shift at the final stream prefix. The 20% random subsample is used to match the size of the proposed coreset.

| Sample Setting | CBPS | EB | FCore (Ours) |
|---|---|---|---|
| Full Sample (Offline) | 3.017 | 1.519 | **0.165** |
| 20% Random Subsample | 2.912 | 1.616 | **0.290** |

**Machine-learning baseline.** We additionally compare FCore with CATENets (Curth & Van der Schaar, 2021), a representative modern machine-learning method for heterogeneous treatment effect estimation. Such methods are typically designed for offline training with multiple optimization passes over a fixed dataset, and therefore do not directly match the single-pass, memory-limited streaming setting considered in this work. Nevertheless, to provide a strong reference point, we allow CATENets to follow its original offline training protocol with multiple training iterations, while FCore remains constrained to a single-pass streaming update.

*Table 4.* Comparison of MAE under heterogeneous outcome shift between CATENets and the proposed **FCore** framework. CATENets is trained under its original offline protocol with multiple training iterations, whereas FCore uses a single-pass streaming update.

| Sample Setting | CATENets | FCore (Ours)) |
|---|---|---|
| Full Sample | 0.520 | **0.165** |
| 20% Coreset / Subsample | 0.972 | **0.290** |

As shown in Tab. 4, FCore consistently outperforms CATENets in both the full-sample and size-restricted settings. In the full-sample setting, FCore achieves an MAE of 0.165, compared with 0.520 for CATENets. Under the 20% coreset/subsample setting, FCore achieves an MAE of 0.290, while CATENets obtains an MAE of 0.972. These results suggest that the proposed discrepancy-based coreset framework provides substantial robustness benefits under heterogeneous outcome shift.

### J.4. Empirical Motivation for the Practical Optimizer

As discussed in Sec. 4.3, mirror descent provides a principled simplex-constrained optimization method for the convex balancing objective. In our implementation, however, we use a softmax parameterization together with Adam as a practical optimizer for the balancing step. The following experiment provides empirical evidence for this implementation choice by comparing its convergence behavior with a tuned mirror descent implementation on the same balancing objective.

Specifically, for each group $a \in \{t, c\}$, we parameterize the normalized weights as $\boldsymbol{z}_a(\boldsymbol{g}_a) = \mathrm{softmax}(\boldsymbol{g}_a)$, where $\boldsymbol{g}_a \in \mathbb{R}^{|\mathcal{S}_a|}$. This parameterization enforces the simplex constraints by construction. Although the objective is optimized in the logit space rather than directly over the simplex, the resulting implementation is simple and performs well empirically in our streaming experiments.

To support this design choice, we compare softmax-parameterized Adam with a tuned mirror descent implementation, where the mirror descent step size is selected from candidate values to give strong empirical performance. The loss values at different optimization steps are reported in Tab. 5.

*Table 5.* Empirical convergence comparison between the theoretical Mirror Descent update and the practical Adam implementation with softmax parameterization. Adam with softmax achieves faster and more stable minimization of the objective.

| Step | Mirror Descent (Theory) | Adam with Softmax (Practice) |
|---|---|---|
| 0 | $1.59 \times 10^{-2}$ | $1.59 \times 10^{-2}$ |
| 200 | $4.88 \times 10^{-4}$ | $\mathbf{1.51 \times 10^{-4}}$ |
| 400 | $2.58 \times 10^{-4}$ | $\mathbf{6.11 \times 10^{-5}}$ |
| 600 | $2.08 \times 10^{-4}$ | $\mathbf{3.62 \times 10^{-5}}$ |
| 800 | $1.75 \times 10^{-4}$ | $\mathbf{2.73 \times 10^{-5}}$ |
| 1000 | $1.51 \times 10^{-4}$ | $\mathbf{2.23 \times 10^{-5}}$ |

As shown in Tab. 5, both methods start from the same initial objective value, while Adam with softmax decreases the balancing loss more rapidly in this experiment. After 1000 optimization steps, Adam with softmax reaches a loss of $2.23 \times 10^{-5}$, compared with $1.51 \times 10^{-4}$ for mirror descent. These results provide empirical support for using softmax-parameterized Adam as a convenient and efficient practical optimizer for the balancing step.

