# OpenReview forum: "Streaming Covariate Balancing via Discrepancy-Based Feature Coresets"
_ICML.cc/2026/Conference — ICML 2026 regular_

### Official Review · Reviewer_cKM2 · 2026-03-10

**Soundness:** 2
**Presentation:** 3
**Significance:** 3
**Originality:** 2
**Overall Recommendation:** 3
**Confidence:** 4

**Summary:**

This paper estimates the average treatment effect (ATE) from streaming observational data under strict memory constraints and potential distributional shifts. The authors propose FCore, a model-agnostic approach that compresses streaming data into discrepancy-based feature coresets using randomized feature representations, enabling covariate balancing over a nonparametric function class with bounded memory. The paper presents theoretical guarantees on approximation error, memory complexity, and estimation error, and empirically demonstrates improved robustness compared with several model-based online causal estimators on synthetic and real datasets.

**Compliance With Llm Reviewing Policy:**

Affirmed.

**Final Justification:**

The rebuttal clarifies several key concerns, particularly the model-agnostic nature of the framework, the distinction from prior coreset methods, and the rationale behind baseline selection in streaming settings. The additional comparison with CATENets and reference to prefix-wise results strengthen the empirical support, and the explanation of the optimization choice helps address the theory–practice gap.

Although I do not fully agree with all of the authors’ arguments, I believe the main concerns have been sufficiently addressed, and I have accordingly raised my score.

**Key Questions For Authors:**

- The theoretical analysis assumes outcome functions lie within an RKHS ball approximated by random features. How realistic is this assumption for real-world datasets, and can diagnostics be provided to assess when it approximately holds?
- The experiments mainly compare against classical estimators (IPTW, G-comp, AIPTW). How does the method perform relative to more recent causal ML methods such as representation learning?
- The streaming setting is emphasized, yet results are mostly reported at the final prefix. Can the authors provide prefix-wise performance curves showing how estimation accuracy evolves over time?
- The optimization used in practice (softmax parameterization with Adam) differs from the mirror descent formulation used in theory. Does this affect convergence guarantees or stability in streaming scenarios?

**Limitations:**

No. The paper includes an impact statement but does not explicitly discuss methodological limitations.

**Strengths And Weaknesses:**

Strengths:
- The paper studies the estimation of causal effects in streaming environments with strict memory constraints and non-stationary data distributions.
- Theoretical analysis is reasonably thorough, providing explicit bounds for memory usage, feature approximation error, and ATE estimation error (e.g., the decomposition of error terms and the final bound in Eq. (13)).
- The proposed algorithm appears computationally efficient, with linear-time updates and polylogarithmic memory growth, which is appropriate for streaming scenarios.
- Empirical results include both synthetic and real-world, demonstrating improved robustness relative to several baseline estimators.

Weaknesses:
- The novelty relative to existing covariate balancing and kernel-based discrepancy methods is somewhat limited. The main contribution appears to be the combination of known techniques (random features, discrepancy-based coresets, and balancing weights).
- The experimental comparison is relatively narrow. The baselines include classical causal estimators (G-computation, IPTW, AIPTW), but omit more recent ML-based causal inference methods (e.g., representation learning), which weakens empirical claims.
- The theoretical analysis relies on several simplifying assumptions (e.g., outcome functions lying in a bounded RKHS ball and random feature approximations), and it is unclear how restrictive these assumptions are in practice.
- The streaming evaluation protocol is somewhat limited. Most experiments report results only at the final prefix, which does not fully demonstrate the advantages of an anytime streaming estimator.
- The optimization procedure used in practice (softmax parameterization with Adam) diverges from the theoretically motivated mirror descent framework, but the implications of this change are not analyzed.

---

> ### Author Rebuttal · Authors · 2026-03-31
>
> **W1**: We respectfully clarify that our work provides several non-trivial contributions that address the fundamental limitations of existing streaming causal inference:
>
> **1. First Model-Agnostic Streaming Causal Inference Method:** Existing streaming causal inference methods (e.g., Luo et al., 2024) rely heavily on parametric models, such as propensity scores. To our knowledge, this is the **first work** to develop a model-agnostic framework for streaming ATE estimation. This architecture is crucial for handling streaming data: the parametric models may suffer from misspecification bias during distribution shifts.
>
> **2. Theoretical Innovation**: We provide a comprehensive theoretical framework that is not a simple extension of coreset theory.
>
> - **Coreset beyond RKHS:** Unlike traditional kernel coresets that require strict RKHS assumptions for construction, our Feature Coreset is designed for the broader integral representation class (Assumption 4.1). We emphasize that our coreset construction is independent of the RKHS ball constraint; the RKHS specialization is utilized only to streamline the analysis of Lemma 4.11 (as explained in line 258).
> - **Principled Error Decomposition:** We establish a rigorous error decomposition specifically for the streaming setting, analyzing how the coreset approximation error and balancing residuals propagate to the final ATE estimate.
>
> **3. Practical Performance:** While many kernel coreset works focus almost exclusively on theoretical bounds, they often lack empirical validation in complex causal tasks. Our work bridges this gap by demonstrating that our coreset-based weighted estimator maintains high precision and stability under distribution shifts.
>
> \
>
> **W2 & Q2**:
>
>  We clarify our baseline selection and provide a rigorous comparison with modern ML methods:
>
> **1. Baseline Selection**
>
> Following the recent benchmark in **streaming causal inference** (Luo et al., 2024), we selected G-computation, IPTW, and AIPTW as primary baselines. These methods are natively compatible with **one-pass streaming updates** to ensure a fair and consistent evaluation within the streaming paradigm.
>
> **2. Comparison with ML methods**
>
> Modern ML methods like representation learning are fundamentally designed for offline settings. Under strict streaming constraints where each sample is processed only once, their performance typically collapses because they require multiple training epochs to converge.
>
> To provide a comprehensive benchmark, we compared our framework against CATENets [1]. To ensure a competitive evaluation for the ML baseline, we allowed CATENets to follow their original offline design with multiple training iterations, while FCore remained strictly constrained to a single-pass update. Despite this offline advantage, the results under Heterogeneous Outcome Shift illustrate our framework's superior robustness:
>
> | **Method**                  | **CATENets** | **Ours (FCore)** |
> | --------------------------- | ------------ | ---------------- |
> | **Full Sample**             | 0.520        | **0.165**        |
> | **20% Coreset / Subsample** | 0.972        | **0.290**        |
>
> [1] Curth, Alicia, and Mihaela Van der Schaar. "Nonparametric estimation of heterogeneous treatment effects: From theory to learning algorithms." International Conference on Artificial Intelligence and Statistics (AISTATS'21)
>
> \
>
> **W3 & Q1**: Please see the response to Reviewer YAgv (W3)
>
> \
>
> **W4 & Q3**: We would like to respectfully clarify that these exact prefix-wise performance curves are **already provided** in Appendix I.2, as **explicitly referenced in Line 395 of the main text**.
>
> \
>
> **W5 & Q4**:
>
> We clarify that the choice of softmax-parameterized Adam over theoretical Mirror Descent is a deliberate design to ensure empirical stability (line 330):
>
> - **Consistency of Optima:** Although this re-parameterization transforms the convex objective into a non-convex one in the logit space, it preserves the Global Optimum.
>
> - **Empirical Comparison**: To validate this architectural choice, we compared the convergence rates of the theoretical **Mirror Descent** versus our practical **Adam (Softmax)** implementation. As shown below, Adam achieves significantly faster and more stable minimization of the loss function:
>
> | **Step** | **Mirror Descent (Theory)** | **Adam with Softmax (Practice)** |
> | -------- | --------------------------- | -------------------------------- |
> | 0        | 1.59e-02                    | 1.59e-02                         |
> | 200      | 4.88e-04                    | **1.51e-04**                     |
> | 400      | 2.58e-04                    | **6.11e-05**                     |
> | 600      | 2.08e-04                    | **3.62e-05**                     |
> | 800      | 1.75e-04                    | **2.73e-05**                     |
> | 1000     | 1.51e-04                    | **2.23e-05**                     |

---

> > ### Author Rebuttal · Reviewer_cKM2 · 2026-04-06
> >
> > I have raised my score, and the reasons are provided in the final justification section.

---

### Official Review · Reviewer_YAgv · 2026-03-10

**Soundness:** 3
**Presentation:** 3
**Significance:** 3
**Originality:** 3
**Overall Recommendation:** 4
**Confidence:** 2

**Summary:**

This paper proposes a model-agnostic method for Average Treatment Effect (ATE) estimation in streaming observational data under memory constraints and distributional shifts. The core idea is to compress streaming data into "Feature Coresets" using random Fourier features and a greedy signing rule that minimizes discrepancy in the feature space. The method achieves O(log n / ε²) memory and O(pdn) time complexity. By directly learning balancing weights without estimating propensity scores, FCore avoids instability under treatment assignment shifts. Theoretical contributions include error decomposition into sampling, coreset approximation, noise, and balancing terms. Experiments on synthetic (heterogeneous outcome shift, Kang-Schafer treatment shift) and real-world (IHDP, COVID-19 vaccine safety) datasets show FCore outperforms online G-comp, IPTW, and AIPTW baselines.

**Compliance With Llm Reviewing Policy:**

Affirmed.

**Final Justification:**

I maintain my current positive score.

**Key Questions For Authors:**

Please see weaknesses.

**Limitations:**

Yes

**Strengths And Weaknesses:**

Strengths

S1. Streaming causal inference under memory constraints and distributional shift is practically important. The vaccine safety surveillance example clearly illustrates why existing model-based online estimators fail under temporal shifts.

S2. By directly learning balancing weights without estimating propensity scores, FCore avoids the instability of IPW under treatment shift. The random feature approximation enables nonparametric balancing over a rich function class.

Weaknesses

W1. Limited comparison with offline balancing methods. How does FCore compare to offline balancing methods (e.g., entropy balancing, CBPS) applied to a random subsample of the same size as the coreset? This would isolate the benefit of the discrepancy-based coreset from the benefit of balancing over model-based methods and quantify the cost of the streaming constraint.

W2. Random feature dimension d is fixed at 200. The error bound depends on d^{-1/4}, but no guidance on how to choose d in practice.

W3. Outcome model assumption (F-class) may be restrictive. Assumption 4.1 requires outcome regressions r_0, r_1 to lie in an RKHS ball. Under outcome shift, post-shift regressions may leave this class, violating the assumption.

---

> ### Author Rebuttal · Authors · 2026-03-31
>
> Thanks a lot for your comments. We present our responses below:
>
> **W1**:
>
> We thank the reviewer for this suggestion. To isolate the benefit of our discrepancy-based coreset, we compared **FCore** against **CBPS** and **Entropy Balancing (EB)** using a 20% random subsample (matching our coreset size).
>
> The results (MAE under Heterogeneous Outcome Shift) at the final stream prefix are as follows:
>
> | **Method**                | **CBPS** | **EB** | **Ours (FCore)** |
> | ------------------------- | -------- | ------ | ---------------- |
> | **Full Sample (Offline)** | 3.017    | 1.519  | **0.165**        |
> | **20% Random Subsample**  | 2.912    | 1.616  | **0.290**        |
>
> \
>
> **W2**:
>
> While the $O(d^{-1/4})$ bound characterizes the asymptotic rate, the practical selection of $d$ is driven by empirical trade-offs. We clarify our choice as follows:
>
> **1. Gap Between Theoretical Rates and Practical Constants:**
>
> As noted in randomized algorithm theory [1], theoretical bounds often reflect worst-case scenarios and omit large constants (e.g., RKHS norms). In practice, approximation errors frequently decay faster than the $O(d^{-1/4})$ rate suggests. Thus, $d$ is typically determined by a **computation-precision balance** rather than a direct calculation from the bound.
>
> **2. Standard Practice and Empirical Robustness:**
>
> Setting $d \in [200, 1000]$ is a standard benchmark heuristic in RFF. Our experiments demonstrate that $d=200$ consistently yields robust and high-precision results across diverse data distributions and streaming scenarios.
>
> **3. Compatibility with Advanced Selection Schemes:**
>
> While our primary focus is the causal framework, our method is fully compatible with advanced RFF optimization techniques to determine $d$ dynamically:
>
> - **Adaptive Sampling:** Data-dependent sampling [2] can improve convergence rates, potentially achieving the same error with a smaller $d$.
> - **Residual Monitoring:** One can adopt an automated estimation scheme by monitoring the Approximation Residual. By pre-defining an error tolerance $\epsilon$, $d$ can be increased until the residual falls below the threshold.
>
> We maintain a fixed $d$ ensures constant memory and time complexity, which is a priority for real-time streaming. We leave the integration of adaptive $d$ selection for future work.
>
>
>
> [1] Rahimi A, Recht B. Random features for large-scale kernel machines[J]. Advances in neural information processing systems, 2007, 20.
>
> [2] Liu F, Huang X, Chen Y, et al. Random features for kernel approximation: A survey on algorithms, theory, and beyond[J]. IEEE Transactions on Pattern Analysis and Machine Intelligence, 2021, 44(10): 7128-7148.
>
> \
>
> **W3**:
>
> We appreciate the reviewer’s critical perspective on the functional space assumptions. We clarify that our framework is designed for graceful degradation under distribution shifts, rather than catastrophic failure, through the following mechanisms:
>
> **1. Theoretical Robustness and Error Decomposition**
>
> While the fixed-radius RKHS ball $\mathcal{H}_R$ is a theoretical simplification, its practical impact is mitigated by the linear decomposition of the error bound. By utilizing universal feature maps (e.g., RFF), the functional class $\mathcal{F}$ is dense in $C(\mathcal{X})$. For any true outcome function $f^\*$, let $f\_R = \text{proj}\_{\mathcal{H}\_R}(f^*)$ be its best approximation within the ball. The total estimation error $\epsilon$ then decomposes as:
>
> $$\epsilon \leq \underbrace{\text{Disc}\_{\mathcal{F}}(P, Q) \cdot R}\_{\text{Estimation Error of } f\_R} + \underbrace{\mathcal{E}\_{approx}(f^*, f\_R)}\_{\text{Approximation Residual}}$$
>
> where $\mathcal{E}\_{approx}$ represents the approximation error. Because the corresponding kernel of our RFF (line 339) is **universal**, this residual remains controlled even for complex functions. The bound does not collapse under distribution shifts; instead, it degrades gracefully and additively.
>
> **2. Empirical Validation under Complex Shifts**
>
> Our experimental design (**Line 955**) explicitly violates the RKHS ball assumption to "stress-test" this robustness. We employ a data stream where the outcome mechanism $b(X)$ incorporates non-smooth and highly non-linear operations (e.g., $\max, \tanh, \sin$, and rational forms). Despite these radical shifts that push the outcome functions far beyond simple smooth functional spaces, FCore maintains a stable and low MAE.

---

> > ### Author Rebuttal · Reviewer_YAgv · 2026-04-01
> >
> > The authors have addressed most of my concerns, and I remain the positive score.

---

> > > ### Author Response · Authors · 2026-04-03
> > >
> > > Thank you very much for your timely response and for your supportive comments!  We sincerely wish you all the best.

---

### Official Review · Reviewer_cswW · 2026-03-13

**Soundness:** 2
**Presentation:** 4
**Significance:** 3
**Originality:** 3
**Overall Recommendation:** 4
**Confidence:** 3

**Summary:**

This paper introduces FCore (Feature Coresets for streaming covariate balancing), a model-agnostic framework designed for Average Treatment Effect (ATE) estimation in streaming data settings. The method addresses two primary challenges in online causal inference: strict memory constraints and distributional shifts in both treatment assignment and outcome-generating processes. The authors leverage discrepancy theory to compress incoming data into mergeable "feature coresets". These coresets utilize a randomized feature map to represent a rich nonparametric function class, allowing covariate balancing to be performed via mean matching in a finite-dimensional feature space. By directly learning balancing weights and bypassing parametric propensity scores, the model gains robustness against non-stationary data environments. The paper provides theoretical convergence guarantees with explicit memory and computational bounds.

**Compliance With Llm Reviewing Policy:**

Affirmed.

**Final Justification:**

The author's response has addressed my concern for the regret theory

**Key Questions For Authors:**

- Given the non-vanishing error floor $(\epsilon + d^{-1/4})$, is there a strategy to adaptively increase $d$ or the bucket size $m$ as $n$ grows to achieve true consistency, and what would be the resulting memory complexity?
-  If a distributional shift moves the outcome regression function $r_a(x)$ outside the assumed RKHS ball, how gracefully does the F-discrepancy bound degrade?
-  How should a practitioner choose the feature-map bandwidth and $d$ in a streaming setting where the "optimal" features might change over time?
- How does the per-step balancing cost $\mathcal{O}(d \log n / \epsilon^2)$ scale when the original covariate dimension $p$ is very large (e.g., $p > 10,000$)?.

**Limitations:**

Yes.

**Strengths And Weaknesses:**

## Strengths
- The theoretical framework is rigorous, providing an error decomposition that accounts for sampling, coreset approximation, noise, and balancing errors.
- The paper is exceptionally clear and well-structured.
- The visualization of coreset fidelity via Kernel Density Estimation (KDE) effectively demonstrates the method's ability to preserve multimodal structures compared to random sketching.
- Combining discrepancy-based coreset construction (specifically greedy feature signing) with causal ATE estimation is a novel and creative application of computational geometry to causal inference.


## Weaknesses
- While the error composition is neat, the estimation is **not consistent**. The second and third terms in the regret bound in Eq. (13) are not diminishing with even infinite observations. They can be dominant terms given large observation datasets.
- The assumption that the outcome regression functions $r_{0}, r_{1}$ stay within a fixed radius-$C$ ball of an RKHS is quite restrictive in a non-stationary "shifting" environment.
- The memory-accuracy trade-off is steep. Because memory usage scales as $\mathcal{O}(d \log n / \epsilon^2)$, achieving a significantly lower error floor requires a memory budget that may negate the benefits of using a coreset approach in the first place.

---

> ### Author Rebuttal · Authors · 2026-03-31
>
> **W1**:
> Actually, the consistency can be proven. Due to space constraints, we provide the proof sketch below.
>
> **1. Restricted Simplex:**
>
> To prevent weight concentration (variance explosion), we optimize over a restricted simplex:
>
> $$W_t(B) := \\{ \mathbf{z}\_t \in \Delta^{|S_t|} : \max_{j \in S_t} z_{t,j} \le \frac{B}{\eta m} \\}$$
>
> where $\eta$ is the strict overlap parameter ($\pi(x) \ge \eta$) and $B \ge 1$ is a constant.
>
> **2. Existence of a Bounded Oracle:**
>
> We construct oracle weights $\bar{\mathbf{z}}\_t$ on the coreset using true inverse-propensity factors: $\bar{z}\_{t,j} \propto v_{t,j}/\pi(x_j)$. By the merge-and-reduce structure, each coreset point $v\_{t,j}$ represents at most one full bucket at some level, implying $v_{t,j} \le n_t/m$. Yields $\bar{z}_{t,j} \le \frac{(n_t/m) \cdot (1/\eta)}{n_t} = \frac{1}{\eta m}$. Thus, $\bar{\mathbf{z}}_t \in W\_t(B)$ is a valid, bounded feasible solution.
>
> **3. Balancing Error Convergence (Second Term):**
>
> The algorithm finds $\hat{\mathbf{z}}$ that minimizes $\mathcal{L}_\Phi$. By optimality:
>
> $$0 \le \mathcal{L}\_\Phi(\hat{\mathbf{z}}) \le \mathcal{L}_\Phi(\bar{\mathbf{z}})$$
>
> As $n, m \to \infty$, the coreset's property Theorem 4.3 ensures that the reweighted coreset measure $\mu_{t,\bar{z}}$ converges to the true population measure. Consequently, $\mathcal{L}\_\Phi(\bar{\mathbf{z}}) \xrightarrow{p} 0$, and by the Squeeze Theorem, the balancing error $\sqrt{\mathcal{L}\_\Phi(\hat{\mathbf{z}})}$ also vanishes.
>
> **4. Variance Term Decay (Third Term):**
>
> In the restricted space $\mathcal{W}_t(B)$, the variance-related term is strictly controlled:
>
> $$\sum_{j \in S_t} \hat{z}\_{t,j}^2 \le \max\_j (\hat{z}\_{t,j}) \le \frac{B}{\eta m} $$
>
> As the bucket size $m \to \infty$, the variance term diminishes.
>
> **Conclusion:**
>
> Both terms vanish as $n, m \to \infty$, confirming the consistency.
>
> \
>
> **Q1**:
>
> True consistency (ensuring the error term $\epsilon + d^{-1/4} \to 0$ as $n \to \infty$) is achievable by adaptively scaling  $m$ and  $d$ with $n$. Based on Theorem 4.3, where $\epsilon \sim \mathcal{O}(1/\sqrt{m})$, this error floor $\mathcal{O}(m^{-1/2} + d^{-1/4})$ vanishes asymptotically by setting $m = d = \log n$. This adaptive coupling results in a memory $\mathcal{O}(dm \log n) = \mathcal{O}(\log^3 n)$.
>
> \
>
> **W2&Q2**: Please see the response to Reviewer YAgv (W3)
>
> \
>
> **Q3**:
>
> **1. Empirical Robustness**
>
> In our experiments, we set $\sigma=3.0$ as a default. Our sensitivity analysis demonstrates that the ATE estimation is remarkably robust to the choice of $\sigma$ across a wide range. As shown below (MAE under Heterogeneous Outcome Shift), the performance remains consistently high and stable within a broad "optimal zone" of $\sigma \in [3.0, 9.0]$:
>
> | Sigma | 1.0  | 2.0  | 3.0  | 4.0  | 5.0  | 6.0  | 7.0  | 8.0  | 9.0  | 10.0 |
> | ----- | ---- | ---- | ---- | ---- | ---- | ---- | ---- | ---- | ---- | ---- |
> | MAE   | 0.74 | 0.23 | 0.18 | 0.15 | 0.13 | 0.11 | 0.10 | 0.10 | 0.11 | 0.12 |
>
> **2. Framework Extensibility**
>
> Our primary contribution is a framework for streaming causal inference. While $d$ and $\sigma$ are fixed in this study, the architecture is inherently extensible to more complex, self-tuning strategies:
>
> - **Adaptive $d$:** The framework can incorporate an automated estimation scheme that monitors the Approximation Residual. By pre-defining an error tolerance $\epsilon$ for the kernel estimation, $d$ can be increased dynamically until the residual falls below the threshold.
> - **Multi-scale $\sigma$:** The framework could employ an **ensemble of RFFs** with multiple bandwidths. This mixture-of-RFFs would allow the model to automatically adapt to varying data densities without manual retuning.
>
> \
>
> **W3**：
>
> We clarify that the $\mathcal{O}(1/\epsilon^2)$ dependence is a **standard theoretical benchmark** in coreset literature, rather than an inefficient "steep" trade-off:
>
> - **Theoretical Lower Bounds:** The quadratic dependence on $1/\epsilon$ is fundamentally unavoidable for some cases. Specifically, for the Gaussian Kernel, [1] proved that when the dimension satisfies $d > 1/\epsilon^2$, the coreset size has a lower bound of **$\Omega(1/\epsilon^2)$**.
> - **Empirical Efficiency:** While theoretical bounds are often conservative due to large constants, the practical trade-off is much smoother. Our experiments performance suggests that the practical "error floor" is reached much earlier than the worst-case theory predicts.
>
> \
>
> **Q4**:
>
> The $\mathcal{O}(d \log n / \epsilon^2)$ balancing cost is **decoupled** from the input dimension $p$. For ultra-high dimensions, the initial feature mapping can be further optimized from $O(pd)$ to $O(d \log p)$ using Fastfood RFF [2], ensuring  computationally tractable.
>
>
>
> [1] Jeff M Phillips et.al. Improved coresets for kernel density estimates. In SIAM 2018, pages 2718–2727.
>
> [2] Quoc Le et.al., “FastFood—approximating kernel expansions in loglinear time,” in ICML, 2013, pp. 244–252.

---

> > ### Author Rebuttal · Reviewer_cswW · 2026-04-03
> >
> > Thank you for your detailed response. My concerns have been largely addressed, and I will adjust my score accordingly after considering the authors' discussions with the other reviewers.

---

> > > ### Author Response · Authors · 2026-04-03
> > >
> > > Thank you for your timely response! We are pleased that our clarifications have addressed your concerns, and we sincerely appreciate your thoughtful evaluation and recognition of our work.

---

### Official Review · Reviewer_jMM9 · 2026-03-16

**Soundness:** 3
**Presentation:** 3
**Significance:** 3
**Originality:** 3
**Overall Recommendation:** 4
**Confidence:** 3

**Summary:**

This paper studies streaming average treatment effect estimation under two challenges: strict memory constraints and distributional shifts in both treatment assignment and outcome generation. The proposed method, FCore, is to compress the data stream into mergeable coresets that preserve the feature mean statistics required for balancing, without the need to estimate propensity scores.

**Compliance With Llm Reviewing Policy:**

Affirmed.

**Final Justification:**

The authors address most of my concerns. I remain the positive score.

**Key Questions For Authors:**

See Weaknesses.

**Limitations:**

Yes.

**Strengths And Weaknesses:**

Strengths:

1. Streaming causal inference is under-explored and very important.

2. The use of mergeable coresets to maintain balancing statistics is interesting. The authors give solid theoretical analysis.

3. The experiments are fairly extensive and show strong empirical performance relative to the selected online baselines.

Weaknesses:

1. The paper defines final weights as $z_t = u_t \odot w_t$, and simultaneously states that these belong to the simplex, which is not true without additional constraints.

2. The analysis assumes a convex optimization over $w_t, w_c$, while the implementation employs optimization over softmax-parameterized logits. Are they equivalent?

3. The paper strongly advertises robustness to distributional shifts (treatment shift and outcome shift), but I don't find very clear theoretical evidence to support the model's robustness to such shifts. Could you please identify them?

---

> ### Author Rebuttal · Authors · 2026-03-31
>
> Thank you for your positive feedback.
>
> **W1**:
> We clarify that $\boldsymbol{z}\_t = \boldsymbol{u}\_t \odot \boldsymbol{w}\_t \in \Delta^{|\mathcal{S}\_t|}$ (line 262) is a standard density ratio reweighting formulation.
>
> - $\boldsymbol{u}\_t \in \Delta^{|\mathcal{S}\_t|}$ is the **base measure** (from the coreset).
> - $\boldsymbol{w}\_t \in \mathbb{R}\_+^{|\mathcal{S}_t|}$ is the **adjustment factor**.
> - $\boldsymbol{z}\_t$ is the **target measure** for ATE.
>
> While $\boldsymbol{w}\_t$ itself is not on the simplex, the optimization ensures $\sum\_j u\_{t,j} w\_{t,j} = 1$, placing $\boldsymbol{z}\_t$ in $\Delta^{|\mathcal{S}\_t|}$.
>
> **Example:** If $\boldsymbol{u}_t = [0.8, 0.2]$ and we seek a balanced $\boldsymbol{z}_t = [0.5, 0.5]$, then $\boldsymbol{w}_t = [0.625, 2.5]$. Here, $\boldsymbol{w}_t \notin \Delta^2$, but the resulting $\boldsymbol{z}_t$ is a valid probability vector.
>
> **W2**:
>
> We clarify the relationship between our theoretical convex analysis and the logit-based implementation as follows:
>
> As shown in Eq. (12) and Proposition 4.12, our objective $\mathcal{L}$ is a quadratic form with positive semi-definite (PSD) Hessians $\mathbf{Q}_t$ and $\mathbf{Q}_c$. This makes $\mathcal{L}$ convex with respect to the adjustment factors $\boldsymbol{w}$. Since the target weights on the simplex are defined as $\boldsymbol{z}_a = \boldsymbol{u}_a \odot \boldsymbol{w}_a$ (where $\boldsymbol{u}_a$ is the fixed base measure), the mapping from $\boldsymbol{w}_a$ to $\boldsymbol{z}_a$ is a linear (affine) transformation, i.e., $\boldsymbol{z}_a = \text{diag}(\boldsymbol{u}_a) \boldsymbol{w}_a$. A fundamental property of convex analysis is that **convexity is preserved** under affine mappings. Therefore, the objective $\mathcal{L}$ remains a convex quadratic function over the probability simplex $\Delta^{|\mathcal{S}_a|}$.
>
> However, we acknowledge that optimizing over softmax-parameterized logits $\mathbf{g} \in \mathbb{R}^m$ is not strictly equivalent to the original convex formulation in $\mathbf{z}$ due to boundary constraints. Nevertheless, several beneficial properties are preserved, such as Global Optimum Preservation: since the softmax function is a surjective (onto) mapping from $\mathbb{R}^d$ to the interior of the simplex $\Delta^d$, the set of global minimizers $\mathbf{z}^*$ of the original convex problem is fully contained within the image of the logit space. We also explicitly state the practical reasons for choosing the softmax method in Line 330.
>
> **W3**:
>
> The robustness of our method is grounded in its **model-agnostic design**:
>
> 1. **Design Motivation:** Unlike parametric methods (e.g., IPTW) that fail under **model misspecification** during shifts, our method directly learns weights via feature balancing. By bypassing functional form assumptions for the propensity score, the model is inherently robust to treatment shifts.
> 2. **Theoretical Evidence:** In Lemma 4.11, the balancing error $e_{\text{bal}}$ is bounded uniformly over the entire nonparametric function class $\mathcal{F}$. As long as the post-shift outcome regressions $r_{a, \text{new}}$ remain within $\mathcal{F}$, the error bound remains valid and unchanged.

---

> > ### Author Rebuttal · Reviewer_jMM9 · 2026-04-01
> >
> > The authors address most of my concerns. I remain the positive score.

---

> > > ### Author Response · Authors · 2026-04-03
> > >
> > > Thanks a lot for your time and positive feedback! We sincerely appreciate your constructive feedback, which has helped us improve the clarity and quality of our work. We are grateful for your careful review and valuable suggestions.

---

### Decision · Program_Chairs · 2026-04-30

**Decision:**

Accept (regular)

**Comment:**

The paper studied the problem of real-time estimation of ATE in streaming observational data. The studied problem is very important and relevant. Most of the concerns are well addressed during rebuttals, so I would like to recommend accept on this paper.